# A Matlab-based toolbox for characterizing behavior of rodents engaged in string-pulling

**Samsoon Inayat\*, Surjeet Singh, Arashk Ghasroddashti, Qandeel, Pramuka Egodage, Ian Q Whishaw, Majid H Mohajerani\***

Canadian Centre for Behavioural Neuroscience, University of Lethbridge, Lethbridge, Canada

**Abstract** String-pulling by rodents is a behavior in which animals make rhythmical body, head, and bilateral forearm as well as skilled hand movements to spontaneously reel in a string. Typical analysis includes kinematic assessment of hand movements done by manually annotating frames. Here, we describe a Matlab-based software that allows whole-body motion characterization using optical flow estimation, descriptive statistics, principal component, and independent component analyses as well as temporal measures of Fano factor, entropy, and Higuchi fractal dimension. Based on image-segmentation and heuristic algorithms for object tracking, the software also allows tracking of body, ears, nose, and forehands for estimation of kinematic parameters such as body length, body angle, head roll, head yaw, head pitch, and path and speed of hand movements. The utility of the task and software is demonstrated by characterizing postural and hand kinematic differences in string-pulling behavior of two strains of mice, C57BL/6 and Swiss Webster.

## Introduction

String-pulling is a proto-tool behavior in which an animal pulls on a string to obtain a reward. Variations of the behavior are observed in over 160 species of animals including birds, bees, rodents and primates and humans (*Jacobs and Osvath, 2015*; *Singh et al., 2019*). The task has been used to assess cognitive function, motivation, brain and spinal cord injury and disease consequences and work/reward relations in animals and as a therapy and exercise tool in humans. If animals are presented with an overhanging string, they adopt a standing or sitting posture and use hand-over-hand movements to reel in the string (*Blackwell et al., 2018a*; *Blackwell et al., 2018b*; *Blackwell et al., 2018c*). Mice will even spontaneously reel in a string, but the behavior becomes more reliable when reinforced by a food reward attached to the string (*Laidre, 2008*). The movement is an on-line act, guided by sensory information from snout vibrissea, and features four hand shaping movements for grasping and releasing the string, and five arm movements for retreiving and advancing the string that are similar in mice, rats, and humans (*Blackwell et al., 2018a*; *Singh et al., 2019*).

The reaching/grasping movements of string-pulling resemble the movements of reach-to-eat movements displayed by rodents in other tasks, including the staircase (*Dunnett, 2010*) and the single-pellet reaching task (*Whishaw, 1996*) but additionally both hands can be assessed concurrently and many reach movements can be collected rapidly. The organized structure of string-pulling makes the task useful for investigating sensory, motor, cognitive and social learning, as well as studying pathophysiology of motor deficits in animal models of motor disorders (*Blackwell et al., 2018c*). The string-pulling task for mice has promise as an unconditioned motor task for phenotyping the growing number of transgenic mouse models of neurological and other diseases but requires an analytical method for its systematic analysis. Although most of the work of advancing the string by a mouse is done with the arms and hands, the movements rely on online sensory control and

**\*For correspondence:**
samsoon.inayat@gmail.com (SI);
mohajerani@uleth.ca (MHM)

**Competing interests:** The authors declare that no competing interests exist.

appropriate postural support. Associated with strain differences or neurological impairment, any of these aspects of behavior may be impaired and compensated for by a variety of sensorimotor adjustments in not just the hands but by the entire body. Thus, the analysis of complex changes in behavior require sophisticated whole-body behavioral analysis in addition to the analysis of skilled hand movements. Currently, most of the analysis is carried out with manual annotation of frames leading to low yield of experiments. Automated analysis of mouse string-pulling is useful way for obtaining an objective kinematic assessment of movements, for identifying the contributions of different body parts to string-advancement, and for gaining insights into inter hand coordination (*Whishaw, 1996*; *Dunnett and Brooks, 2018*). The objective of the present paper is to describe a state-of-the-art method for analyzing mouse string-pulling.

We present a Matlab-based toolbox to facilitate the analyses of video data of different strains of intact or neurologically compromised mice engaged in string pulling. The toolbox features a graphical user interface for video data analysis and follows the analysis framework with two major components, a global assessment of whole-body position and speed and fine kinematic assessment of movements of body, ears, nose, and hands (*Figure 1*). As a first step, the user can run optical flow analysis on video data (image sequence) to estimate velocity vector fields and speed frames. Next using descriptive statistics and temporal measures of Fano factor, entropy, and Higuchi fractal dimension, the user can characterize whole-body position, speeds, and their temporal variability. With principal and independent component analyses of the image sequence and speed frames, the user can assess global position and speed patterns to observe regions of significant motion. Additional spatial measures of entropy, sharpness, and Hausdorff fractal dimension allow the user to quantify randomness, edges, and geometrical complexity respectively and use these quantifications to statistically compare groups of animals. We demonstrate that with the whole-body measures mentioned above, we can detect overall differences in position and motion patterns of two different strains of mice, C57BL/6 (Black) and Swiss Webster Albino (White). Furthermore, using color-based image segmentation and heuristic algorithms, the software allows detailed automatic tracking of body, ears, nose, and hands for kinematic analyses (*Video 1* and *Video 2*). The software also allows the user to manually curate and correct automatic tagging of body parts. Finally, the user can either utilize Matlab scripts provided with the software for comparison of groups of mice or export data to Microsoft Excel for further flexible data analyses. A modular approach has been used to build the toolbox, allowing users to add additional analytical probes.

## Results

### Characterization of position with central-tendency descriptive statistics of image sequence shows larger number of distinct positions for white compared to black mice

Position characterization of a mouse during the string-pulling behavior (representative frames shown in *Figure 2A and C* for Black and White mice, respectively) can be grossly done using central-tendency descriptive statistics that is mean, median, and mode, of the image sequence. By observing the mean, median, and mode frames (*Figure 2B and D*), one can infer the average to most frequent positions of the mouse during string-pulling. These frames of the representative Black and White mice show sitting and standing positions, respectively. For statistical comparison of these frames between groups of Black (N = 5) and White (N = 5) mice, spatial entropy, sharpness, and Hausdorff fractal dimension (HFD) measures were calculated and compared using Student's t-tests. Spatial entropy is measured from 1D projection of the frames while sharpness and HFD operates directly on a 2D frame. Spatial entropy, sharpness, and HFD quantify randomness, edges, and complexity of a frame, respectively. The measures were similar for both groups except that the spatial entropy of the mean frames was larger for White compared to Black mice (p=0.0092, effect size = 2.157, *Figure 2E*). This suggests that the mean frame of White mice had more random values as compared to Black mice owing to larger number of distinct positions.

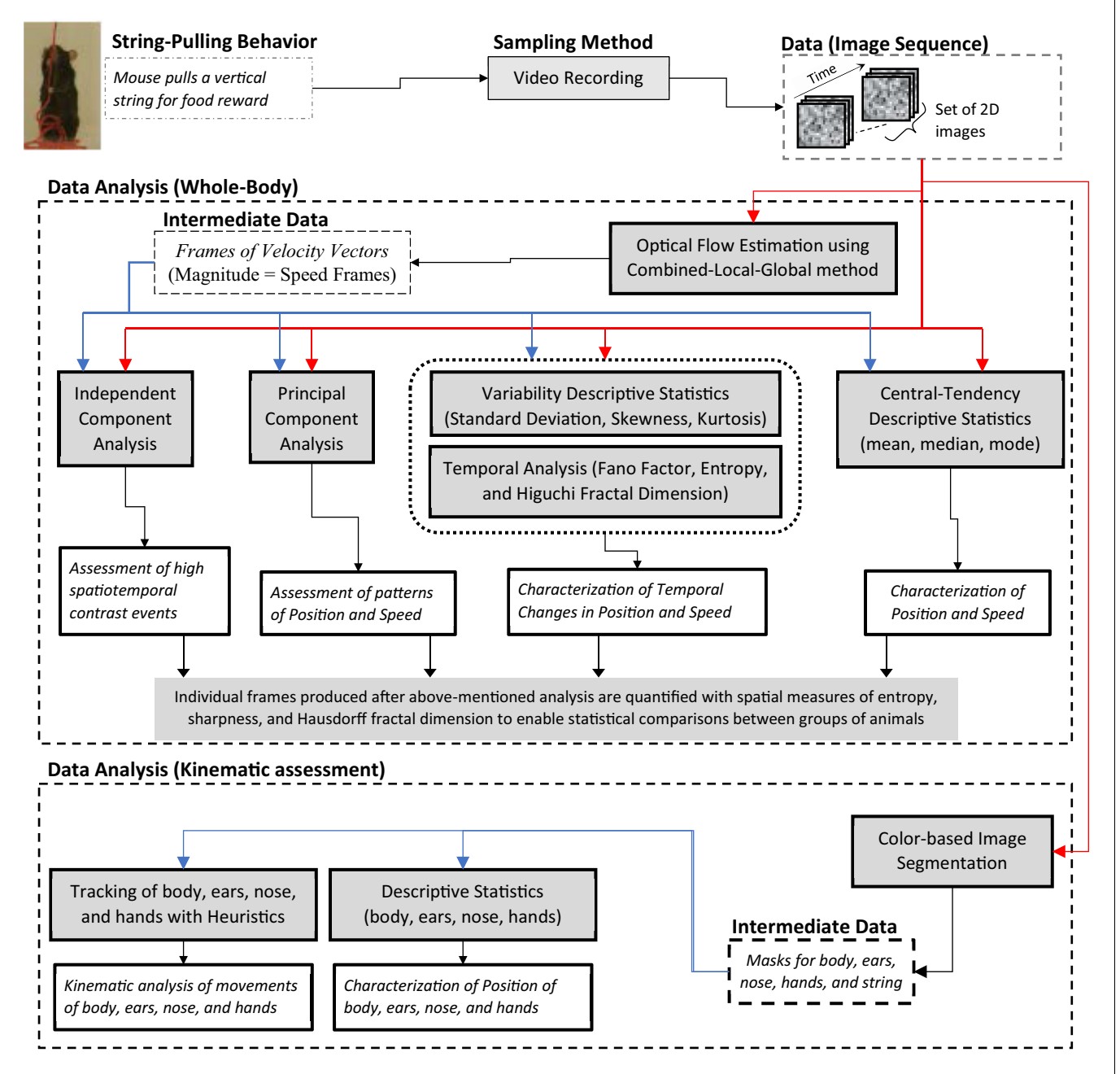

**Figure 1.** String-pulling video analysis framework. Characterization of whole-body position and speed and their temporal variability is assessed with descriptive statistics, Fano factor, entropy, and Higuchi fractal dimension. Speed is estimated with optical flow estimation. Principal and independent component analysis of frames of image sequence and speed frames provide overall assessment of position and speed patterns. Measures of spatial entropy, sharpness, and Hausdorff fractal dimension on frames obtained in earlier steps are used to statistically compare global position and motion patterns of groups of animals. For fine kinematic assessment of the motion of body, ears, nose, and hands, they are tracked using image segmentation and heuristic algorithms. Shaded boxes represent methods.

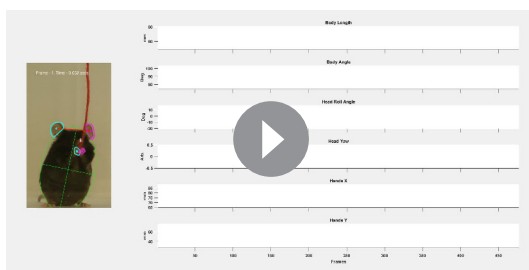

**Video 1.** Video of the representative. Black mouse pulling string overlaid with tracked body, ears, nose, and hands also showing frame number and time of frames. Temporal progression of the measured parameters, body length, body angle, head roll angle, head yaw, the X and Y positions of hands from the lower left corner (origin) in the original frames.
https://elifesciences.org/articles/54540#video1

## Characterization of temporal changes in position with variability descriptive statistics, Fano factor, entropy, and Higuchi fractal dimension of image sequence depicts larger variability of positions of White compared to Black mice

To assess the temporal changes in the position of the mouse and the string, we found measures of the time series of individual pixels including variability descriptive statistics (standard deviation, skewness, and kurtosis), Fano factor, temporal entropy, and Higuchi fractal dimension (*Figure 3*). The standard deviation frame shows a summary of spatial variation of the mouse position over time for example how the body of the representative Black mouse was in a sitting as well as standing position (*Figure 3A*). Conversely, the White mouse remained in a standing position and moved its head left and right that is yaw motion (*Figure 3B*). The skewness and kurtosis frames conveyed similar information and captured rare events such as positions of the string in relation to the mouse body. Both skewness and kurtosis had large values around the mouse body. Like the standard deviation frame, the Fano factor frame also showed change in position over time but with respect to the mean position. Fano factor frame highlighted the changes more than the standard deviation frame; for example for both representative Black and White mice, there is large variation at the top of the head region where the ears are. The temporal entropy of the image sequence quantified randomness of the time series of individual pixels and hence could also be translated to the change in position that is quantity of motion at the individual pixel location. For the representative Black mouse, it is largest around the boundary of the mouse (note the thin red layer near body edge in the Entropy frame). For the lower half of the body, the thickness of the red band, signifying value, is thin whereas it is large in the upper half of the body where it captured the change in mouse's sitting to standing posture (*Figure 3A*). For the White mouse, the temporal entropy value was quite large within the mouse's body depicting large motion of the body. In the upper half of the body, it also revealed a yaw motion of the head. The Higuchi fractal dimension (HiFD) quantified the complexity of the time series of individual pixels in the image sequence. For the representative Black and White mice, HiFD captured changing positions of the string with larger values (hot regions) outside the mouse body. Within the mouse body, it also captured the position of hands which were colocalized with the string in most of the frames (at least one of the hands). The smaller HiFD values (colder regions) for the Black mouse depict the gradual transition in mouse positions from sitting to standing while for the White mouse, they depict its head's yaw motion.

To assess temporal changes in positions of Black versus White mice, we compared the distributions of values of the above-mentioned parameters. The values of parameters were obtained from their respective frames (e.g. values of HiFD from the HiFD frame) using a mask obtained from the mean frame (see Materials and methods) depicting average location of the mouse. The outline of this mask for the representative Black mouse is shown in the mean frame in *Figure 2B* (for White mouse in *Figure 2D*). For each animal, the probability density function (PDF) of the values of a parameter were estimated using Gaussian Kernel density estimation. The cumulative density function

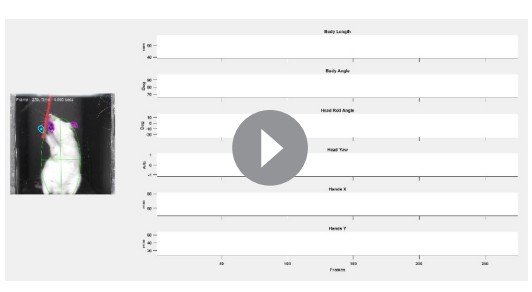

**Video 2.** Same as *Video 1* but for White mouse.
https://elifesciences.org/articles/54540#video2

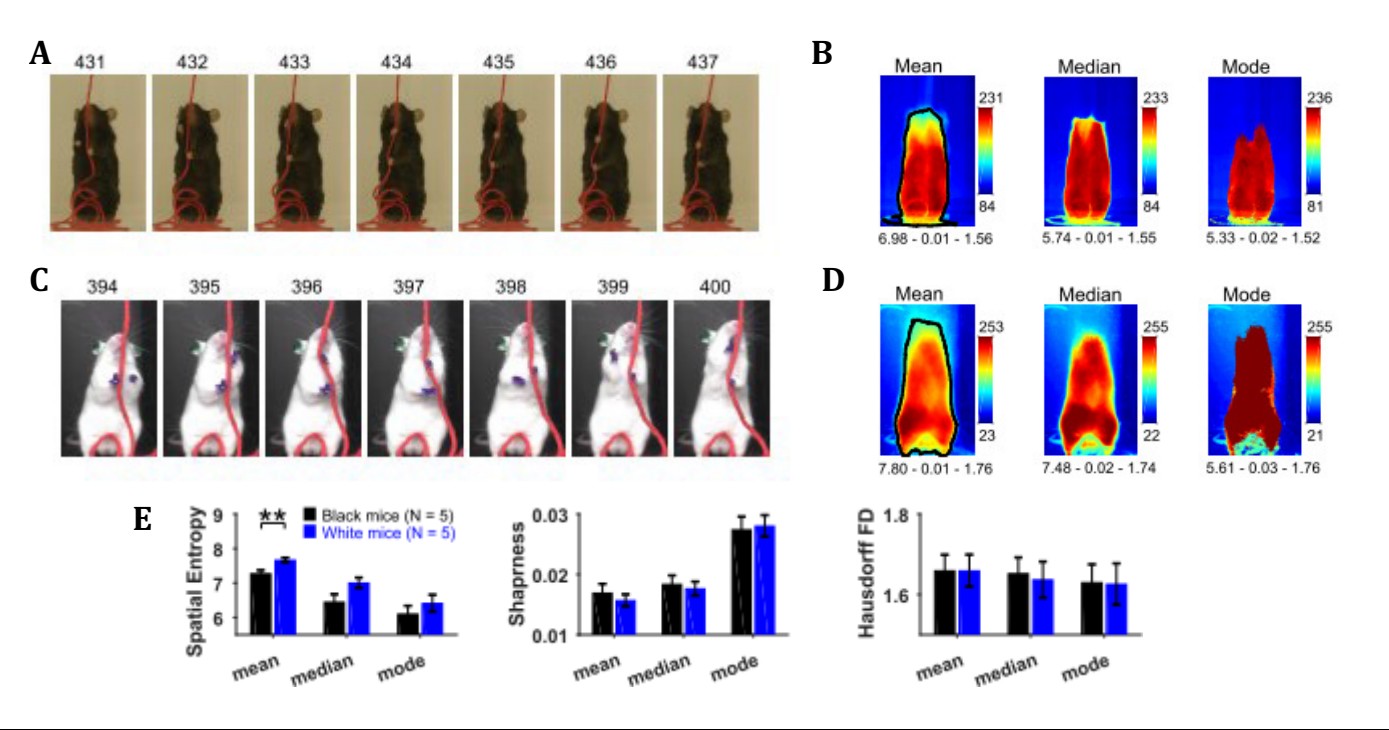

**Figure 2.** Characterization of whole-body position using central-tendency descriptive statistics. (**A**) Representative subset of image frames of a string-pulling epoch of a Black mouse. (**B**) Central-tendency descriptive statistics: mean, median, and mode of string-pulling image sequence for a Black mouse. Mean, median, and mode frames represent average to most frequent position of mouse. Shown below each frame are the respective values of measures of spatial entropy, sharpness, and Hausdorff fractal dimension separated by hyphens. The black line in the mean frame shows the boundary of a mask generated by selecting all pixels with values above the average value of the mean frame. (**C**) and (**D**) same as (**A**) and (**B**) respectively but for a White mouse. Descriptive statistics of representative Black mouse show sitting as most frequent posture while that of the White mouse is upright standing. (**E**) Mean ± SEM spatial entropy, sharpness, and Hausdorff fractal dimension shown for frames of descriptive statistics from 5 Black and 5 White mice.

(CDF) was then determined from PDF. The Mean ± SEM CDFs from 5 Black and 5 White mice are shown in *Figure 3C–H*: standard deviation, skewness, kurtosis, Fano factor, temporal entropy, and HiFD respectively. Alongside the CDF curves, means over animals are also shown that were obtained by first finding a mean (single value for each animal) of the distribution of a parameter. The means of skewness, temporal entropy, and HiFD were significantly larger for White compared to Black mice (individual Student's t-tests, $p < 0.001$, $p = 0.008$, and $p = 0.007$ with the following effect sizes: 4.035, 2.219, and 2.737, respectively). The opposite was the case for Kurtosis values ($p = 0.009$, effect size = 2.138). Together, these significant differences highlight the differences underlying dynamics of pixel intensities that depict changes in positions of Black and White mice for example more negative skewness and smaller temporal entropy and HiFD values for Black mice show more stable and gradual changes in positions.

We also calculated the spatial measures of entropy, sharpness, and Hausdorff fractal dimension for all the individual frames depicting variability and then used individual Student's t-tests (*Figure 3I*). Note that these measures are for the whole frames in contrast with above-mentioned analysis where values of parameters for example temporal entropy and HiFD were extracted using a mask representing average position of mouse. Thus, these measures take into account motion of string along with that of mouse. The measures of spatial entropy and sharpness for the standard deviation frames were significantly larger for the White compared to Black mice ($p = 0.023$ and $p = 0.017$ with the following effect sizes: 1.825 and 1.910, respectively *Figure 3I*). The spatial entropy of HiFD frame was significantly larger for the Black compared to White mice ($p = 0.023$, effect size = 1.776). Finally, the sharpness of the Fano factor, temporal entropy, and HiFD frames was significantly larger for the Black compared to White mice ($p = 0.007$, $p < 0.001$, and $p = 0.007$ with the

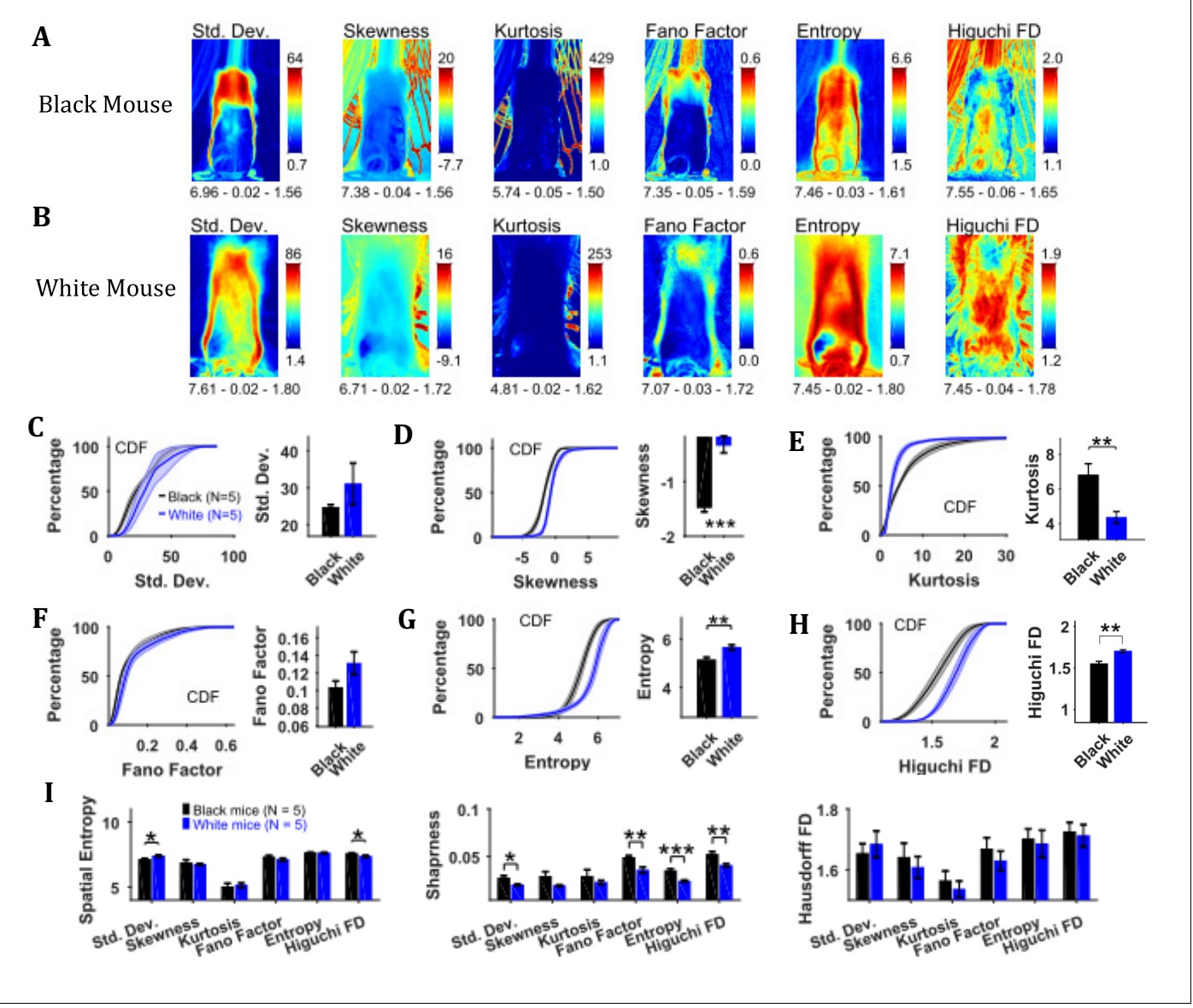

**Figure 3.** Characterization of changes in position with variability descriptive statistics, Fano factor, entropy, and Higuchi fractal dimension. (A) Representative variability descriptive statistics – standard deviation (Std. Dev.), skewness, and kurtosis, Fano factor, Entropy, and Higuchi fractal dimension of string-pulling image sequence of a Black mouse. Standard deviation depicts summary of spatial variation over time (mouse posture from sitting to standing). Skewness and kurtosis capture regions of rare events around the mouse body such as position of string. Shown below each frame are the respective values of spatial entropy (SE), sharpness (S), and Hausdorff fractal dimension (HFD) separated with a hyphen. (B) same as (A) but for White mouse. The White mouse has greater yaw and pitch head motion as seen in the standard deviation. (C) Mean ± SEM cumulative distribution function (CDF) of the standard deviation values within a mask obtained for each individual mouse from their respective mean frames. For the representative Black and White mice, outline of these masks is shown in *Figure 2B and D*, respectively. (D–H) Same as (C) but for other parameters. (I) Mean ± SEM SE, S, and HFD shown for frames of parameters mentioned above.

following effect size: 2.261, 3.305, 2.580, respectively). Combined these results suggest larger changes in positions of White compared to Black mice.

## Characterization of speeds with central-tendency descriptive statistics reveals larger speeds of White compared to Black mice

By using the CLG optical flow algorithm, velocity vector fields were determined for each consecutive pair of image frames and then speed frames were obtained by finding the magnitude of velocity

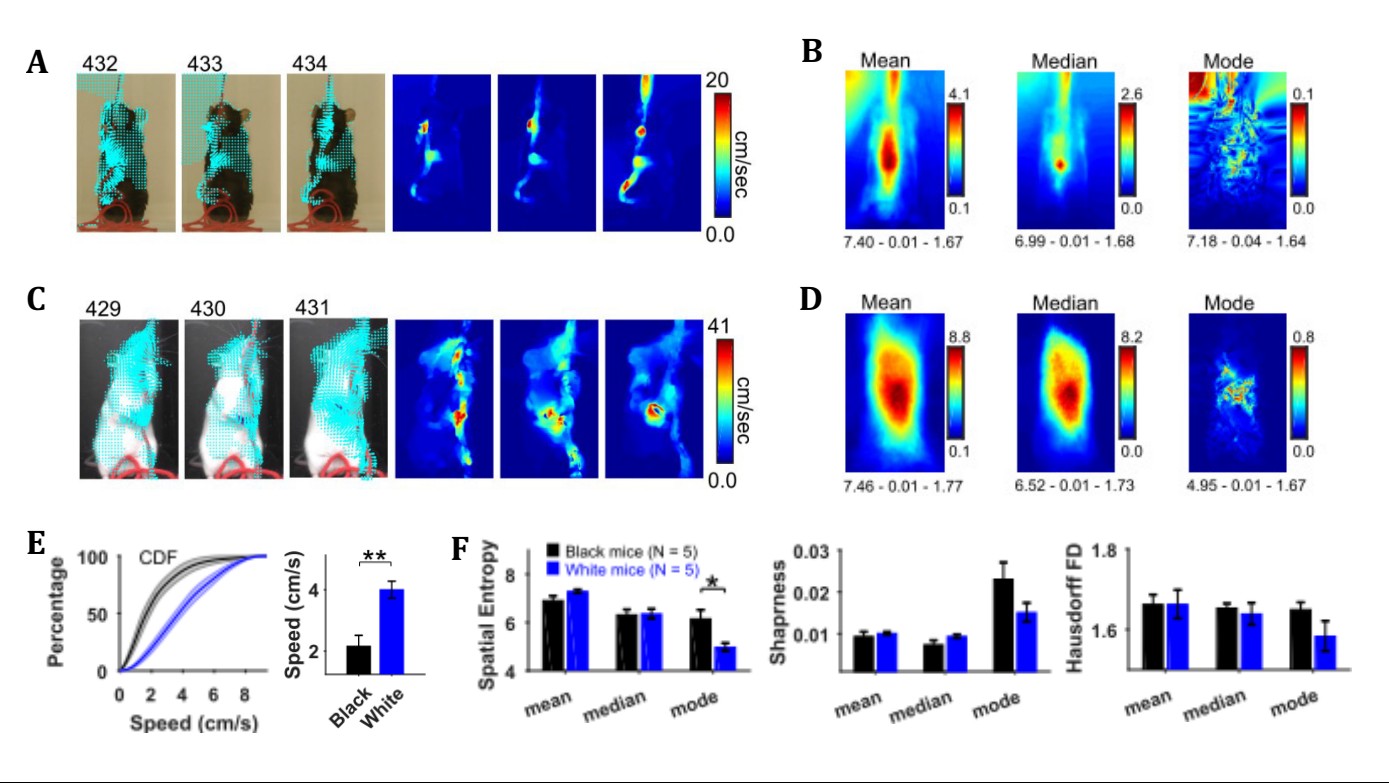

**Figure 4.** Characterization of speed with central-tendency descriptive statistics reveals White mice have larger speeds compared to Black mice. (**A**) Output of the optical flow analysis overlayed on three representative image frames. Velocity vector is shown for every fifth pixel (along both dimensions). For a pixel, the magnitude of the velocity vector shows instantaneous speed (cm/s). For the same image frames, respective speed frames are shown on the right. (**B**) Descriptive statistics of the speed frames. Shown below each frame are the respective values of spatial entropy (SE), sharpness (S), and Hausdorff fractal dimension (HFD) separated with a hyphen. The unit of color code is cm/sec. (**C**) and (**D**) same as (**A**) and (**B**) respectively but for White mouse. (**E**) Average cumulative distributions of speeds from the mean speed frames of Black (N = 5) and White (N = 5) mice. Shaded regions show SEM. Bar graph show Mean ± SEM over animals of mean speeds. (**F**) Mean ± SEM SE, S, and HFD shown for frames of descriptive statistics from 5 Black and 5 White mice.

vectors. The algorithm faithfully captured the motion dynamics of video frames (*Figure 4A and C*). The mean, median, and mode speed frames exhibited the spatial distribution of average to most frequent speeds (*Figure 4B and D*). Observation of the respective frames for representative Black and White mice show clear similarities as well as differences. For example, the peak values of mean and median frames for the White mouse are larger than those for the Black mouse but peak location is similar for both, and close to the center of mouse's body where the hands release the string. Similar conclusions can be derived from the median and the mode frames.

To compare the distributions of speeds for Black and White mice, speed values were chosen from the mean speed frames using a mask (depicting the average position of mouse, see outline in the mean frames in *Figure 2*). The mean speed frame was used because the number of frames in a string-pulling epoch can be different from mouse to mouse. Therefore, normalization is done for comparison purpose. The mask was used to get a better estimate of mouse speeds rather than those of spurious speeds of background pixels. The average cumulative distribution of speeds for White and Black mice is shown in *Figure 4E*. The mean speed for White mice was significantly larger than that for Black mice (Student's t-test, p=0.003, effect size = 2.607). Comparison of White and Black mice was also done with measures of spatial entropy, sharpness, and Hausdorff fractal dimension for the mean, median, and mode frames. These measures were not different for groups of Black and White mice except that the spatial entropy of mode frames was significantly larger for Black mice compared to White (Student's t-test, p=0.017 and effect size = 1.894) highlighting that for White mice most frequent speeds are those of hands positioned around the center of body. Overall,

these results suggest that White mice had faster movements than Black mice more localized on their body.

## Characterization of temporal changes in speed with variability descriptive statistics, Fano factor, entropy, and Higuchi fractal dimension of speed frames shows larger variability of speeds of White compared to Black mice

Changes in speed were characterized by using variability descriptive statistics, Fano factor, entropy, and Higuchi fractal dimension (HiFD) of speed Frames (*Figure 5*) in a similar manner as shown above for image sequence. For both representative Black and White mice, the standard deviation frame showed the largest variation in speeds around the center of body where the hands release the string. For the Black mouse, variation in speeds can also be seen in the upper left corner where the string appeared and moved in a subset of frames. For the White mouse, large variation can also be seen around the head depicting its yaw motion. The skewness and kurtosis frames show large values around the mouse body showing the distribution of speeds is more non-normal around the mouse (same as for the image sequence). The Fano factor frames measured dispersion of speeds in time relative to the mean or noise-to-signal ratio. For both White and Black mice, the dispersion appears to be greater around the mouse body. The temporal entropy was greater within the mouse body as compared to outside and vice versa for HiFD.

To assess the temporal changes in speed for groups of Black and White mice, we compared the values of above-mentioned parameters using distributions of values obtained from each animal using its respective mask calculated from the mean frame of image sequence (see outline in mean frame in *Figure 2B and D* for representative Black and White mice). The Mean ± SEM cumulative distributions are shown in *Figure 5C–H* along with Mean ± SEM over animals for standard deviation, skewness, kurtosis, Fano factor, temporal entropy, and HiFD. The mean skewness (positive) and Fano factor were larger for Black compared to White mice (Student's t-test, p=0.028 and p=0.023 with the following effect sizes: 1.686 and 1.781, respectively) owing to smaller means (see representative mean frames in *Figure 4*). The opposite was true for temporal entropy and HiFD measures (p=0.003 and p=0.008 with the following effect sizes: 2.700 and 2.495, respecctively). The spatial measures of entropy (spatial), sharpness, and Hausdorff fractal dimension for the frames of above-mentioned parameters were not significantly different between the Black and the White mice (*Figure 5I*); however, they have been included here for the sake of demonstration of analysis. Some of the measures likely would become significant with larger sample size. Taken together, these results suggest larger variability, temporal randomness, and temporal complexity of speeds within the mouse body for White vs Black mice.

## Principal component analysis of image sequence and speed frames for assessment of spatial patterns of position and speed

Principal component analysis of the image sequence and speed frames revealed orthogonal components representing regions with the largest variance in intensity and speeds, respectively (*Figure 6*). The first principal component (PC1) of the image sequence showed the most dominant position of the mouse and was similar to frames obtained by first-order descriptive statistics. For the Black mouse, PC2 mostly captured the difference between sitting and standing posture of the mouse. Higher order components captured other dominant features of images for example, positions of head and string. PC1 of the speed frames mostly represented motion of the hands, which constituted significant motion. Motion of the string body, head, and ears appeared in higher principal components. For the Black mouse, PC1 of speed also captured dominant speed of the string in the upper left-hand corner. Since, PC1 captures the largest variance, we compared the characteristics of PC1 for groups of Black and White mice. For PC1s of both image sequence and speed frames, the spatial entropy, sharpness, and Hausdorff fractal dimensions of Black and White mice were similar (Student's t-test, *Figure 6E*), except the spatial entropy of PC1 of image sequence was larger for White compared to Black mice (p=0.011, effect size = 2.095). However, the mean explained variance of PC1 of the image sequence for Black mice was significantly larger than that of White mice compared with a Student's t-test (p<0.0001, effect size = 6.363), whereas the explained variance of PC1 of the speed frames was smaller for Black mice compared to White mice (p=0.032, effect

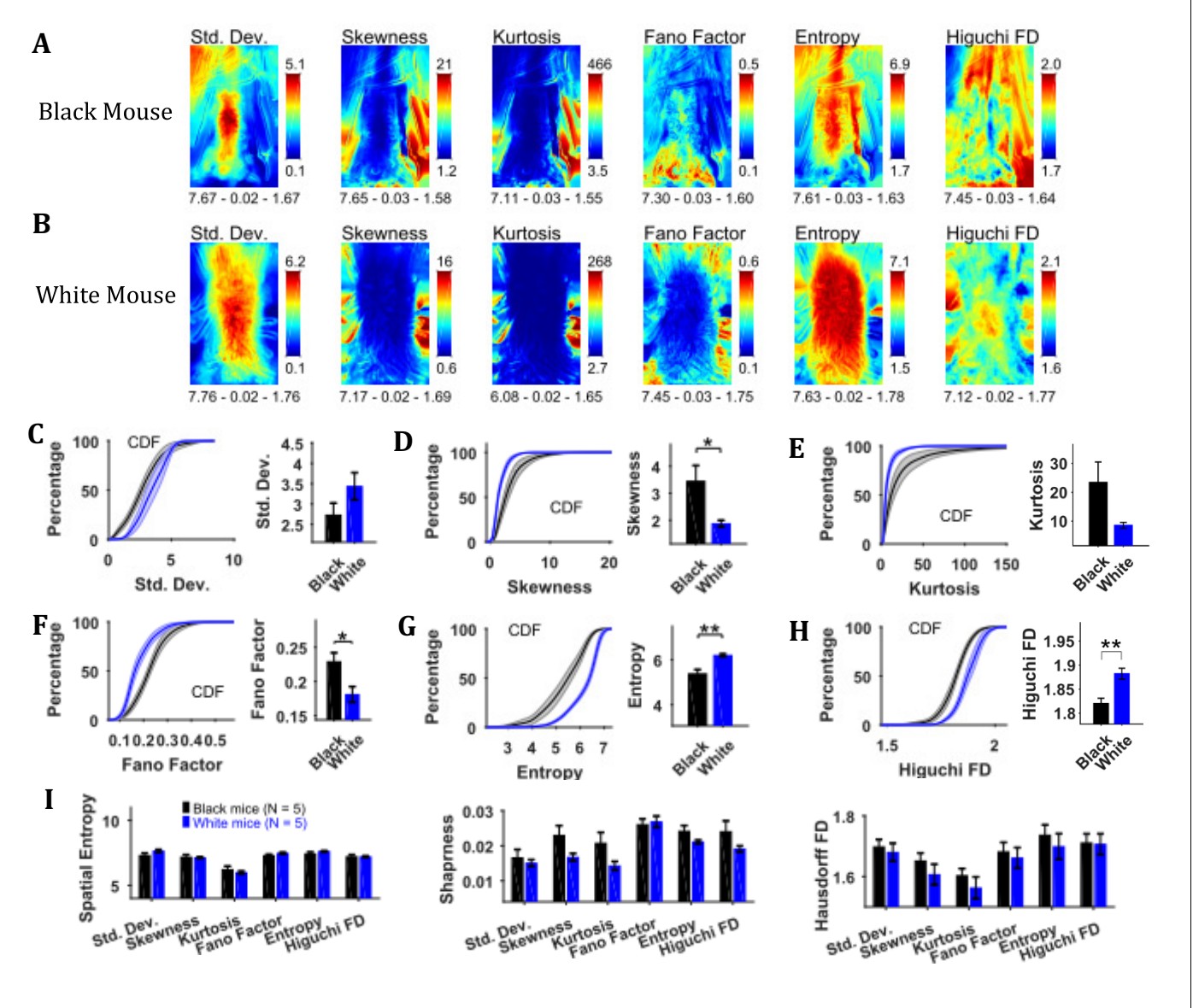

**Figure 5.** Characterization of changes in speed with variability descriptive statistics, Fano factor, entropy, and Higuchi fractal dimension. (A) Representative variability descriptive statistics – standard deviation (Std. Dev.), skewness, and kurtosis, Fano factor, Entropy, and Higuchi fractal dimension of speed frames of a Black Mouse. Shown below each frame are the respective values of spatial entropy (SE), sharpness (S), and Hausdorff fractal dimension (HFD) separated with a hyphen. (B) Same as (A) but for a White mouse. (C) Mean ± SEM cumulative distributions (CDF) of standard deviation values within a mask obtained for each individual mouse from their respective mean frames. For the representative Black and White mice, outline of these masks is shown in *Figure 2B and D*, respectively. (D–H) Same as (C) but for other parameters. (I) Mean ± SEM SE, S, and HFD shown for frames of parameters mentioned above.

size = 1.636). These findings corroborate results reported above that White mice have larger variability of positions and higher speeds compared to Black mice.

## Independent component analysis of image sequence and speed frames for detecting events with high spatiotemporal contrast

Independent component analysis (ICA) allows extraction of statistically independent information within a data set by decomposing it into non-Gaussian components. For the string-pulling motion sequence, the most dynamic movements are those of the hands and the string and they contribute to the non-gaussianity of the data set. When applied directly on the image sequence, ICA extracts

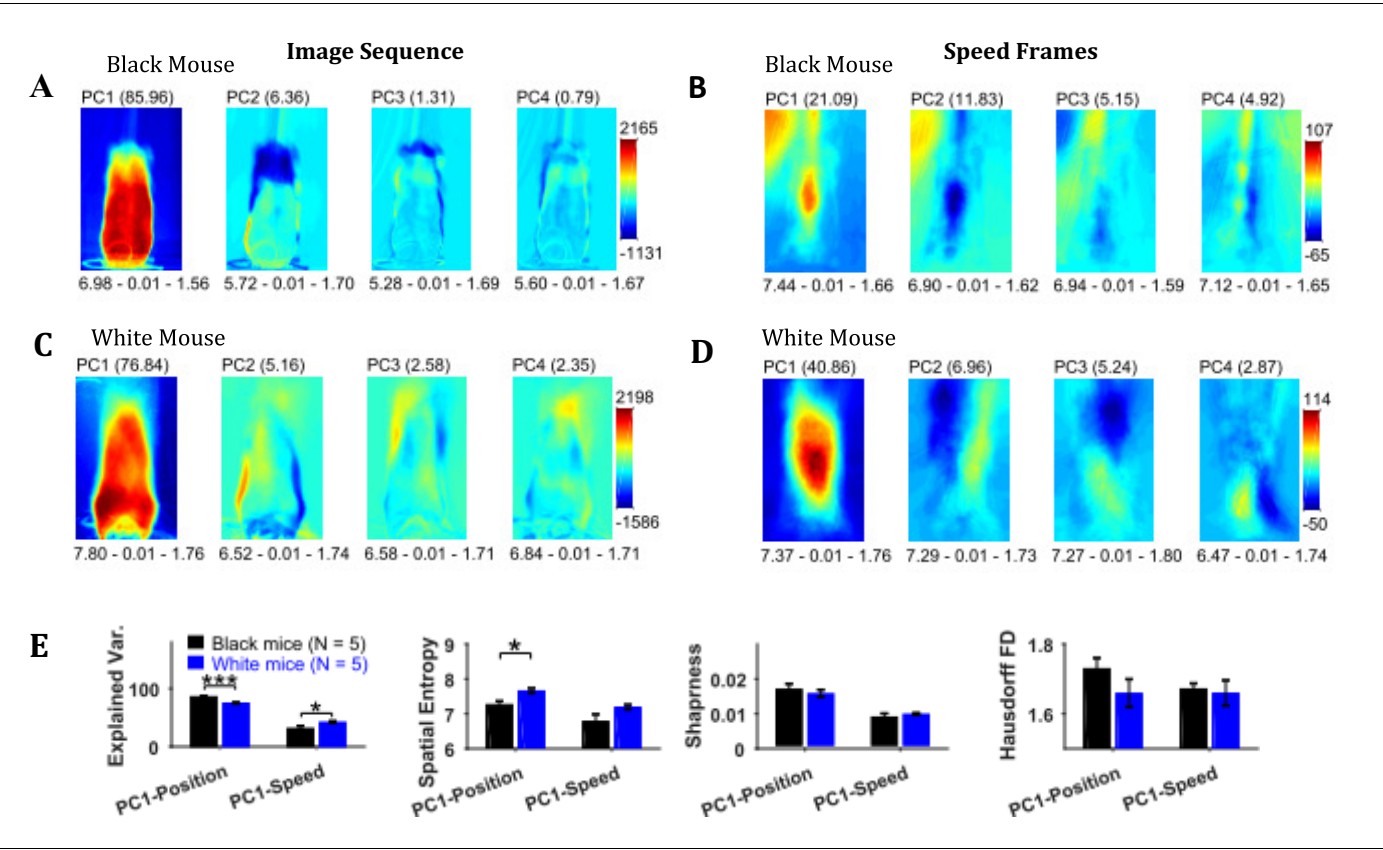

**Figure 6.** Principal component analysis of image sequence and speed frames for assessment of patterns of positions and speeds. (**A**) The first four principal components (PC) of the image sequence of the representative Black mouse. The number in parentheses indicates the value of explained variance. Shown below each frame are the respective values of spatial entropy (SE), sharpness (**S**), and Hausdorff fractal dimension (HFD) separated with a hyphen. (**B**) First four PCs of speed frames for the Black mouse. (**C**) and (**D**) same as (**A**) and (**B**) respectively but for the representative White mouse. (**E**) Mean ± SEM values of explained variance (Var.), SE, S, and HFD for PC1 of positions and speeds.

events that have high spatiotemporal contrast that is where there are sudden changes in an image. Sudden changes include the appearance and disappearance of a hand or string in a frame. *Figure 7A and C* show representative independent components (ICs) of image sequences for the representative Black and White mice, respectively. The heat maps show small regions with red and dark blue colors identifying regions of high spatiotemporal contrast (red and blue colors for positive and negative values respectively indicate appearance and disappearance of objects in frames). The ICs of speed frames were similar in appearance to the ICs of image sequence as they picked events with high spatiotemporal changes in speeds that is mostly movements of the hands and the string. In order to obtain a global picture of the independent components over time, maximum (Max) and minimum (Min) projections were determined (*Figure 7B and D*) for ICs of both image sequence and speed frames. For both White and Black mice, Max and Min projections of independent components show regions where the hands and string were located. In order to validate these observations, ICA was applied to a synthetic mouse data set in which only the hands moved, and this revealed the prominent appearance of hand locations in Max and Min projections of independent components (*Figure 7E and F*). ICA was also done on the principal components of image sequence and speed frames obtained earlier. These analyses yielded results and conclusions similar to those for ICA on the image sequence and speed frames. For the Max and Min projections of ICs, spatial entropy, sharpness, and Hausdorff fractal dimension measures were determined and compared for groups of Black and White mice. In comparisons where differences were significant (with Student's t-test) are shown in *Figure 7G*. The spatial entropy as well as sharpness was larger for White mice compared to Black supporting results reported above (p-values for comparison of spatial entropy of Max-IC-Position and Max-IC-Speed are 0.012 and 0.011 with the following effect sizes: 2.043 and 2.091,

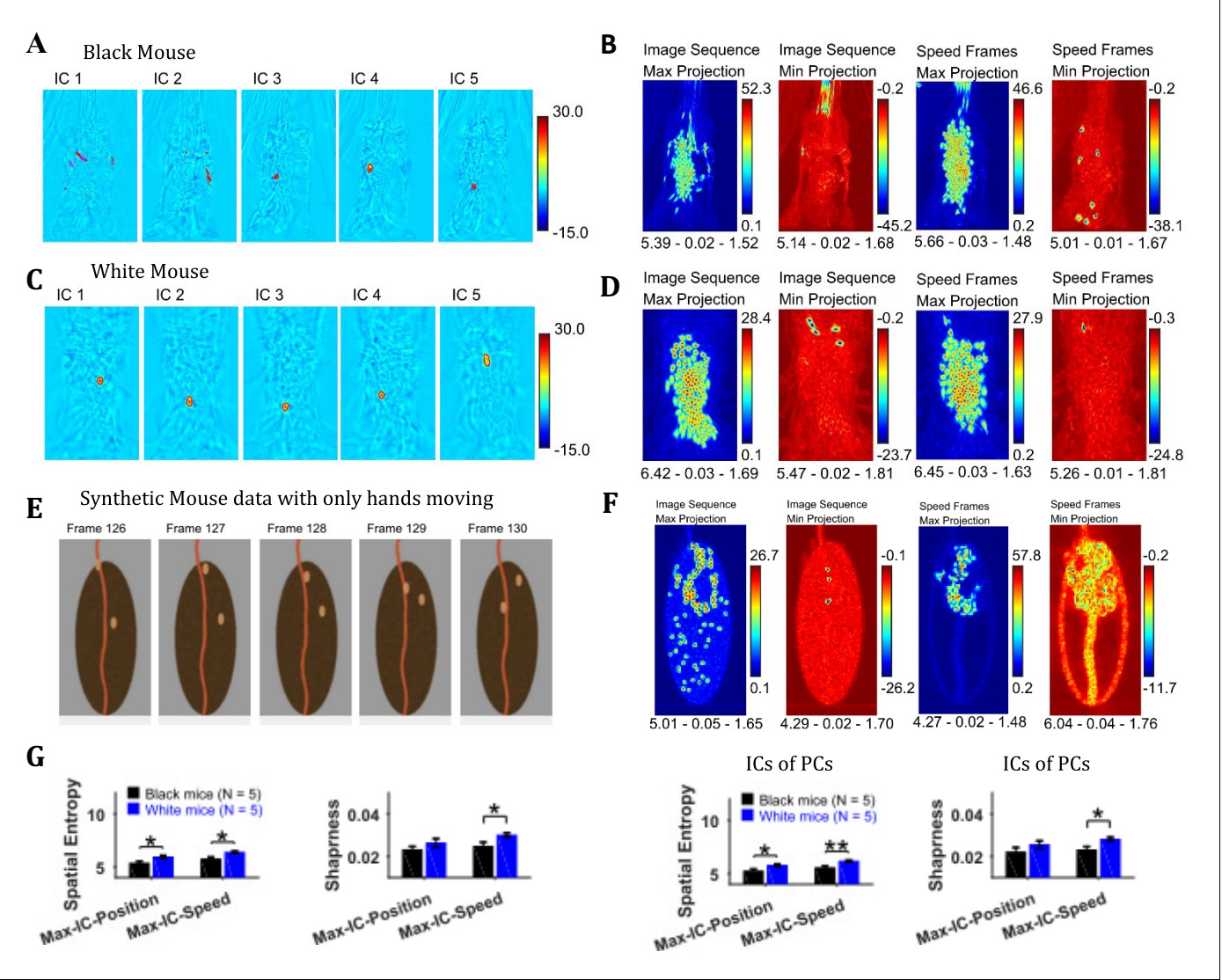

**Figure 7.** Independent component analysis captures regions with high spatiotemporal contrast. (**A**) Representative independent components for Black mouse. Individual independent components mostly represent snapshots of position of the hands and strings, as they are most dynamic in the image sequence. (**B**) Maximum (Max) and minimum (Min) value projections of all independent components of image sequence and speed frames. Max and Min projections grossly represent locations of sudden changes happening in an image sequence for example, hand appearing or disappearing at a new location. Shown below each frame are the respective values of spatial entropy, sharpness, and Hausdorff fractal dimension separated with a hyphen. (**C**) and (**D**) similar to (**A**) and (**B**) respectively but for White mouse. (**E**) Representative image frames of synthetic mouse data in which only hands moved. (**F**) Similar to (**B**) but for synthetic data. (**G**) Mean ± SEM of spatial entropy and sharpness of Max projections of ICs for image sequence, speed frames and their respective PCs.

respectively; p-value for comparison of sharpness of Max-IC-Speed is 0.034 with effect size = 1.618, for IC frames of PCs, p-values for comparison of spatial entropy of Max-IC-Position and Max-IC-Speed are 0.030 and 0.008 with the following effect sizes:1.661 and 2.204, respectively; p-value for comparison of sharpness of Max-IC-Speed is 0.025 with effect size = 1.741).

## Descriptive statistics on image sequences of masks of body, ears, nose, and hands

Descriptive statistics on the masks' image sequence can reveal useful information about the extent of motion of individual body objects, for example, the standard deviation of body, ears, nose, and

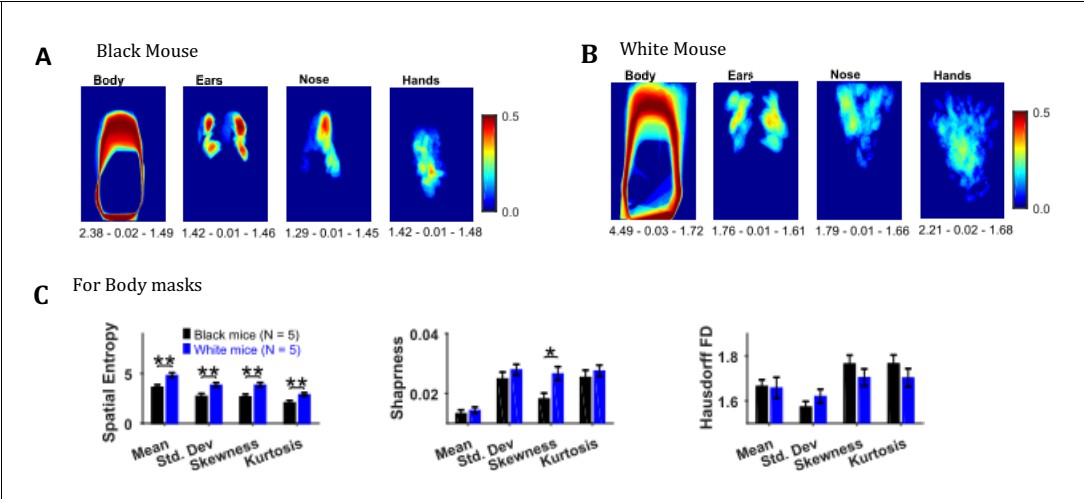

**Figure 8.** Image segmentation based on object colors for finding object masks and gross assessment from descriptive statistics of masks. (**A–B**) Standard deviation of the masks image sequence of body, ears, nose, and hands for overall assessment of respective motion patterns of Black and White mice, respectively. Shown below each frame are the respective values of spatial entropy (SE), sharpness (S), and Hausdorff fractal dimension (HFD) separated with a hyphen. (**C**) Mean ± SEM SE, S, and HFD shown for Body masks.

hand masks as shown in *Figure 8A and B* for representative Black and White mice provides the variability in the position of respective objects. One can easily see individual differences between the representative Black and White mice for example the body positions of a Black mouse are distinct as seen by crisp standard deviation frame compared to a White mouse which has diffuse appearance. Comparing the spatial measures between groups of Black and White mice revealed significantly higher spatial entropy for mean, standard deviation, skewness, and kurtosis frames for White mice (p=0.006, p=0.008, p=0.006, p=0.009 respectively with the following respective effect sizes; 2.356, 2.198, 2.370, and 2.132, *Figure 8C*). The sharpness of skewness was also larger for White mice (p=0.022, effect size = 1.785). The spatial measures were not different for ears, nose, and hands.

## Identification of fur, ears, and nose for quantification of body and head posture of Black mice

Body length, body tilt, head roll, head yaw, and head pitch are estimated from the masks for fur, ears, and nose (*Figure 9*). From the mask for fur, the mouse's body centroid and body fit are estimated. Since in the fur mask, segregated regions may be segmented because of an overlapping object on the body such as the string in *Figure 9* A1, a convex hull (Matlab function bwconvhull) operation is applied to combine regions and find a cumulative area representing fur (*Figure 9* A2, also see *Figure 9—figure supplement 1*). Matlab function 'regionprops' is then used to identify regions in the cumulative mask and their properties for example area, centroid, orientation, major and minor axis length of an ellipse fitted on the region (*Figure 9* A7, green). Major axis length and orientation of the ellipse are then used as estimates of body length and tilt, respectively. In the example string pulling epoch in the sample video (lasting about eight secs), the starting body length is about 40 mm and increases by ~1.5% as the mouse stands and elongates its body to virtually walk up on the string (*Figure 9B*). The body tilt however remains vertical (between 80°−90°) throughout the epoch (*Figure 9C*).

To estimate head yaw, roll, and pitch, regions representing the right and left ears and the nose are first identified using the respective masks and the elliptical fit for the body found earlier. An elliptical mask is generated for the body using the elliptical fit and the intersection of an inverse of the elliptical mask (converting 0's to 1's and vice versa) with that of ears to provide regions that are outside the ellipse for body (*Figure 9* A4). If segmented regions are close, they are combined (*Figure 9—figure supplement 1*) and very small regions are ignored. In data for the Black mouse, only two regions remained after the end of this procedure and they were easily labeled as the right and

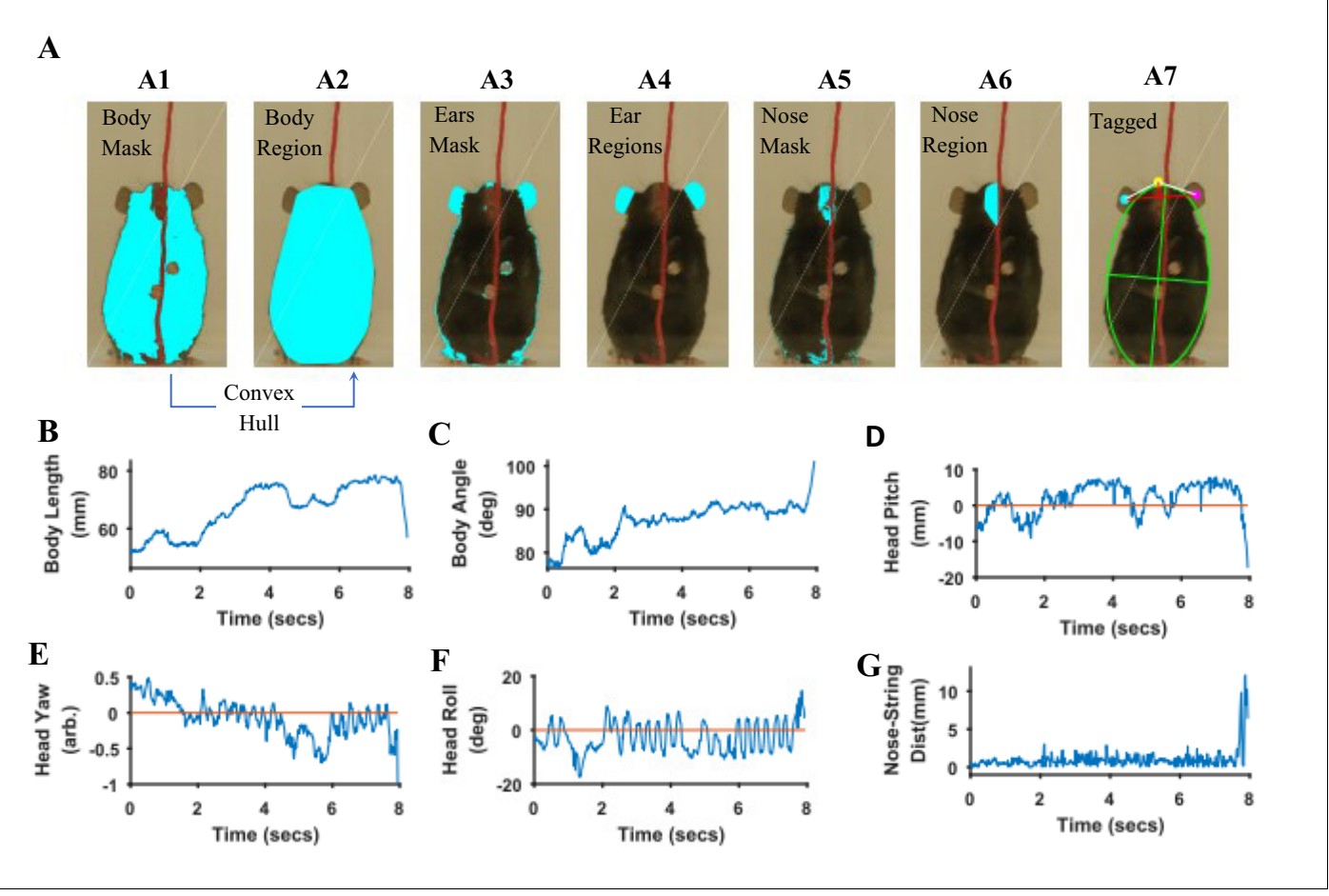

**Figure 9.** Quantification of body and head posture. (A) Tagging of body, ears, and nose. From masks found with fur color (A1), body region (A2) is identified and fitted with an ellipse (A7, green). From the ears' mask (A3), ear regions are identified by using body centroid position found earlier (A4, cyan and magenta dots in A7). Nose mask (A5) is used to identify the nose region (A6), which is fitted with an ellipse to identify nose position (A7, yellow dot). (B) Body length vs time. Body length is the length of major axis of the body fitted ellipse. (C) Body Angle vs time. Body angle is that of the major axis of the body fitted ellipse from horizontal. (D) Head Pitch vs time. Head pitch is estimated as the perpendicular distance between the nose and a straight line joining the ears (small red line in A7). Positive and negative values correspond to the nose above or below the line respectively. (E) Head Yaw vs time. Head yaw is estimated by finding the log of the ratio of distance of right ear from the nose to the distance of left ear from the nose (ratio of lengths of White lines in A7). A value of 0 indicates 0 yaw angle. (F) Head Roll Angle vs time. Head roll is the angle of the line joining ear centroids from horizontal using the right ear as the origin. (G). Shortest distance of the nose from the string vs time.

The online version of this article includes the following figure supplement(s) for figure 9:

**Figure supplement 1.** Algorithm to reduce the number of regions by spatial clustering.

left ears by finding their centroid location relative to body centroid that is regions left and right to the body centroid would be right and left ears respectively. However, if multiple regions remain after the intersection procedure, regions overlapping (or closest) to ear regions in a previous frame are chosen. Spatial clustering is also applied to combine fragmented regions that are close together. The centroids of the identified ear regions are used to represent their location (*Figure 9* A7, cyan and magenta dots). Next the nose is identified from the nose mask (*Figure 9* A5). After identifying spatial regions, clustering is first used to combine regions that are close together (*Figure 9—figure supplement 1*) and then the region closest to the top of the body ellipse is labeled as the nose region (*Figure 9* A6). An ellipse is fitted onto the nose region and the top of this ellipse is used as a single point to represent nose location (*Figure 9* A7, yellow dot). Head yaw is then estimated as the log of the ratio of distance between the nose and the right ear to the distance between nose and left ear. A head yaw value of 0 indicates 0 head yaw angle which means that the face of mouse is facing the camera. In the representative string-pulling epoch of the Black mouse, head yaw values are

initially greater than 0 and gradually reduce to values less than 0, indicating that the mouse is moving its head from left to right (*Figure 9E*). Head roll angle is estimated by finding the angle of line joining the centroids of the regions representing the right and left ears from the horizontal (*Figure 9* A7, long red line). While the mouse pulls the string, the head rolls at a frequency of about three rolls/s (*Figure 9F*) in the sample video (*Video 1*). However, there seems to be a bias for this mouse to keep its head titled such that the right ear is slightly higher than the left ear. Head pitch is estimated as the perpendicular distance between the nose and line joining the two ear centroids (*Figure 9* A7, small red line). Positive and negative values of head pitch correspond to nose position above or below the line joining the ears, respectively. In the sample video, the head pitch stays positive most of the time, as the mouse is moving its nose up for sniffing or using its whiskers to track the string (*Figure 9D*). Furthermore, as has been previously shown (*Blackwell et al., 2018a*), the nose stays close to the string (*Figure 9G*).

## Identification of right and left hands and kinematic measures of hand movement of Black mice

Since both hands move rapidly and have similar colors, it is a challenge to distinctly identify the right and left hands without using a heuristic-based approach. To make automatic identification simpler, we let the user initialize the location of hands in the starting frame, the first frame of an epoch, where a user manually marks two regions representing right and left hands, respectively. In the subsequent frames, automatic identification of the hands in a frame is achieved by using information of the hand location in the previous frame. Automatic identification uses masks for the hands and finds regions using 'regionprops' function of Matlab. As a first step, the number of regions is reduced such that regions that most likely would represent the hands are kept; for example regions with very small areas (less than 20 mm$^2$) are neglected (*Figure 10* A2 represent selected regions after extra regions were neglected). Close by regions are also combined using spatial clustering (*Figure 9—figure supplement 1*). Once a set of reduced number of regions is obtained, the algorithm (*Figure 10—figure supplement 1*) identifies the left and right hands. If only one region remains, the algorithm assumes that the hands are touching each other. It divides the region into two equal regions and labels the subregions as left and right hands based on their distances from left- and right-hand regions in the previous frame (*Figure 10—figure supplement 1*, *Figure 10* A3 and A4). If two distinct regions remain, the same algorithm is used. If more than two regions are identified, regions are further reduced based on their distances from (and overlaps between) left- and right-hand regions in the previous frame and the regions are fed to the same algorithm in a recursive manner.

The regions representing hands were also determined by first finding maximally stable extremal regions (MSER) in the original frame (*Obdrzalek and Mikulik, 2009*). Using the Matlab function 'detectMSERFeatures' MSERs were found (*Figure 10B1*). Color definition for the hands was then used to find MSERs that contained hand colors and a mask was thus generated (*Figure 10B2*). For the representative Black mouse video, hand masks generated this way were in general better than those created with the 'rangesearch' function (see Materials and methods); that is they had lesser number of spurious regions. These masks were then treated as described above to find regions representing the left and right hands (*Figure 10B3 and B4*). Both methods of finding hand regions are made available in the graphical user interface using choices that can be made with check boxes.

Since the automatic algorithm is not deterministic but rather makes estimates using heuristics, it is possible that the algorithm incorrectly labels regions as either the right or left hand. In this case, the user can stop the automatic process and manually choose the appropriate identification in an individual frame. The automatic identification of hands in a frame depends upon the position of hands in the previous frame, therefore the user might have to reprocess all subsequent frames.

Once the positions of the hands are identified in all frames in behavioral epochs of interest that is the x and y coordinates of centroids are found. They can be used to calculate kinematic measures. For example, paths of hand motions can be visualized or the vertical and horizontal positions of each hand from an origin point (lower left corner of the original frame) can be determined and plotted (*Figure 10C and D*). In the representative epoch of string pulling for the Black mouse, the stroke length of pulling is initially shorter and gradually grows for both right and left hands. With respect to the common origin, for the vertical motion, the two hands are alternating, whereas for the horizontal motion, the two hands move to the right and left together. From the position plots, the speed of

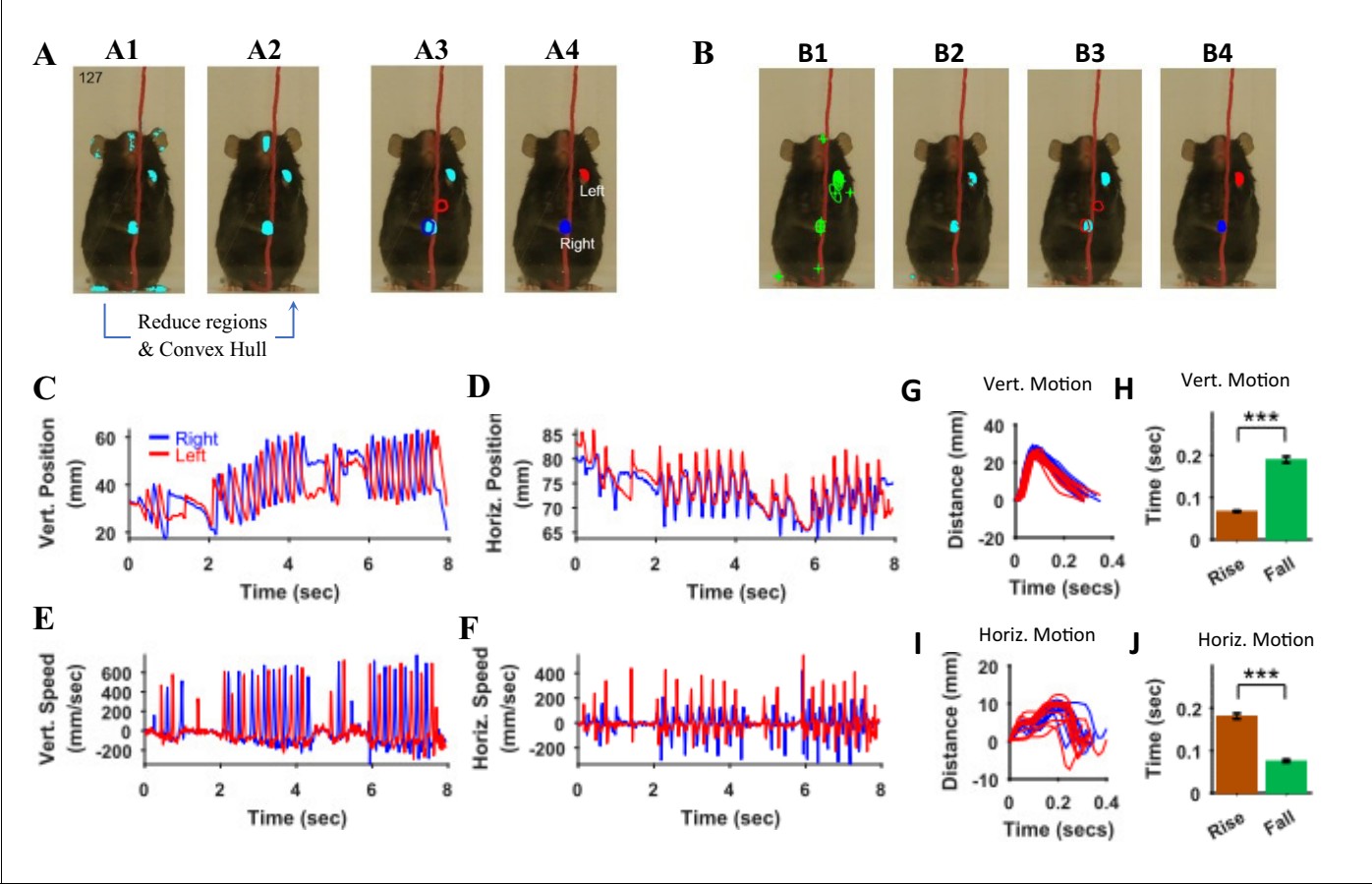

**Figure 10.** Identification of hands and kinematic measures. (**A**) Representative frame (127) with overlayed regions in hand masks (**A1**). Regions in A1 are reduced to find regions that would most likely represent the hands (**A2**). Note that it includes a region in the nose area. Using the position of hands in the previous frame (blue and red outlines in A3), the hands are identified in the current frame (**A4**). (**B**) same as in A but masks are first found by identifying maximally stable extremal regions in the original image (**B1**). (**C**), (**D**) Vertical (Vert) and Horizontal (Horz) positions vs time of right (blue) and left (red) hands, respectively. Positions are measured from the lower left corner of frame as the origin. Where horizontal movements are synchronized, vertical movements of right and left hands are alternating. (**E**), (**F**) Vertical and horizontal speed vs time of right and left hands. Maximum vertical speed is about three times larger than maximum horizontal speed. (**G**) Distance vs. time for vertical motion plotted from all reach cycles (lift-advance-grasp-pull-push-release). Red for left hand and blue for right hand motion. (**H**) Mean ± SEM rise and fall times for vertical motion reaches. (**I**) and (**J**) similar to G) and H) respectively but for horizontal motion. Also see *Video 1* and *Video 2*.

The online version of this article includes the following figure supplement(s) for figure 10:

**Figure supplement 1.** Algorithm to label right- and left-hand regions from two unlabeled regions.

the hands can be found by finding a time derivative. The vertical speed of the hands is about two to three times faster than the horizontal speed. During the whole epoch, there are two bouts of rapid pulling each about two secs long (from 2 to 4 s and from 6 to 8 s). During these bouts, the path shapes of strokes of one hand were similar to each other and to the strokes of the other hand (*Figure 10G*). For each stroke, the rise time that is motion from point of release of the string (near the bottom) to the point of the grasp (near the top) is smaller than the fall time which is the time mouse is pulling the string downwards (Student's t-test, p<0.001, N = 21 cycles, *Figure 10H*). For the horizontal motion, however, the strokes are bimodal (*Figure 10I*), and the rise time is larger than the fall time (Student's t-test, p<0.001, N = 22 cycles, *Figure 10J*).

## Quantification of body and head posture and kinematic analyses of White mice and comparison with Black mice

Distinguishing between the fur and other body parts (hands and ears) was difficult for the White mice, resulting in the generation of poor-quality masks. This was because of lower contrast between

body parts and not due to other factors such as lighting conditions. Therefore, using sharpie markers, we painted the hands and ears of the mice with blue and green colors respectively (*Figure 11* A1). After painting, image segmentation for identifying the ears and the hands was easier (*Figure 11* A3 and A6 respectively). All topographical and kinematic parameters for head posture and motion of the hands were determined (*Figure 11B–E*) in a similar fashion as described above. Whereas the Black mouse gradually stood up, increasing its body length while pulling, the White mouse moved the whole body in a rhythmic fashion up and down (*Figure 11B*). The White mouse occasionally used its mouth to hold the string (*Video 2*).

There was no significant difference between maximum body length, minimum body length, and change in body length (max – min) between White and Black mice, Student's t-test, p>0.05. However, body linear and angular speeds were significantly larger for White compared to Black mice (p=0.0198 and p=0.0199 with the following effect sizes: 1.837 and 1.834, respectively, *Figure 11F*). The dominant frequency in body length oscillations (determined with Fourier transform of the time series of body length) was not different between Black and White mice. Similarly, the dominant frequency of motion for hands was not different. The mean amplitude of both hands was smaller for White mice while mean vertical speed was larger, but neither was statistically significant (*Figure 11G*, shown for right hand only). The peak left-hand vertical speed, however, was slightly

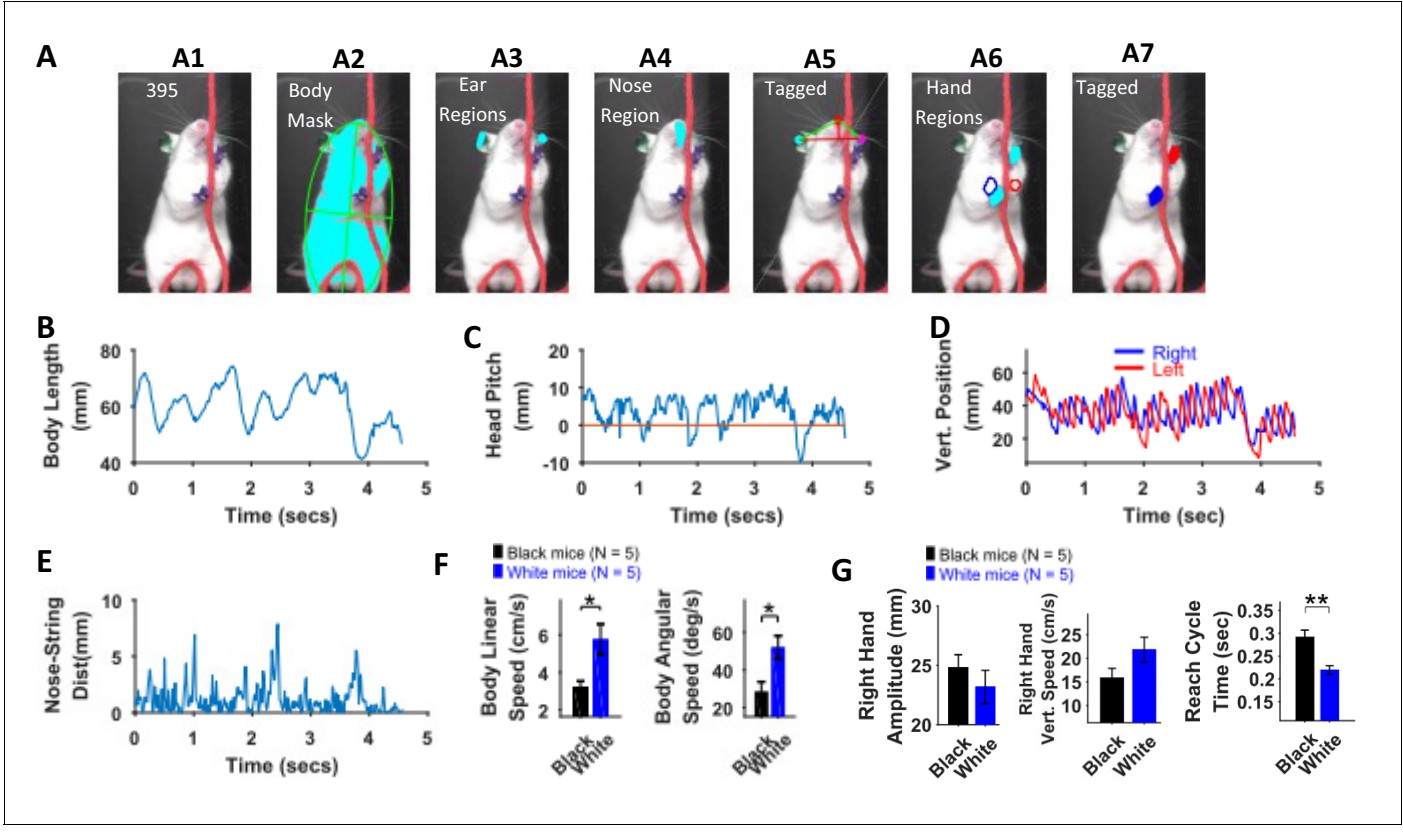

**Figure 11.** Identification of body parts and quantification of kinematic measures in White mice. (A) Procedure for tagging of body, ears, nose, and hands was like that used for Black mice except that the ears and hands were colored green and blue respectively. From masks found using fur color (A2), the body region is identified and fitted with an ellipse. Ear and nose regions (A3 and A4, respectively) are identified and used to measure head posture (A5). Hand regions (A6) are identified based on the position of the hands in the previous frame (A6, red and blue boundaries), the position of the hands in the current frame are identified (A7). (B) Body length vs time. (C) Head Pitch vs time. In frames where both ears are not visible, y-distance between nose and ear is measured as head pitch. D) Vertical position of the hands (from bottom of frame) vs time. (E) Shortest distance of the nose from the string vs time. (F) Mean ± SEM of body linear and angular speeds from five Black and five White mice. (G) Mean ± SEM amplitude, vertical speed, and time period of reach cycle of right hand.

The online version of this article includes the following figure supplement(s) for figure 11:

**Figure supplement 1.** Temporal profiling of reach cycle for black and white mice with Dynamic Time Warping.

significantly larger for White mice (p=0.0445, effect size = 1.506). Average time of right-hand reach cycles (see examples in *Figure 10G*) characterized as lift-advance-grasp-pull-push-release (*Blackwell et al., 2018a*) was significantly smaller for White compared to Black mice (p=0.005, effect size = 2.46) corroborating previous finding that white mice have higher speeds (*Figure 11G*). Time warping was also used to find an average reach cycle individually for each animal followed by another time warping operation to find group average (*Figure 11—figure supplement 1*). The amplitudes of time-warped reach cycles were similar for Black and White mice.

## Benchmarking of algorithms for finding body parts

To determine the accuracy of heuristic algorithms presented here, the percentage of frames in which body parts were correctly detected automatically (verified with manual observation) is calculated. To display the accuracy of tagging of body parts for the data set being analyzed, a push button in the GUI has been provided. *Table 1* shows accuracy results for data from five black and five white mice. The overall average accuracy for all body parts is ~96%, with higher accuracies for fur and ears compared to those for nose and hands. This is because the nose is sometimes hidden behind the string, making detection difficult for the algorithm while the hands are moving at a rapid rate and the algorithm cannot differentiate between regions in the mask that belong to hands versus those that belong to hand-like body parts for example nose.

## Neural networks for identifying ears, nose, and hands

We also trained neural networks to analyze string-pulling videos to detect ears, nose, and hands in Black mice. We used the Python based frame work of Deeplabcut toolbox (*Mathis et al., 2018*; *Nath et al., 2018*) to train the ResNet50 network for identifying ears, nose, and hands (*He et al., 2016*). Four separate networks were trained; three for the individual recognition of the ears, nose, and hands and one for the combined recognition of all three in a single step. Since the first three networks for individual recognition had better performance (visually observed), they were used in subsequent analysis. Details of training the network and using it to analyze videos can be found in *Mathis et al., 2018*. For training each neuronal network, 54 frames were labelled (18 from three different videos) and used to create the training dataset. For each network, training took about 36 hr on a GPU (NVIDIA GeForce GTX 1080 with 8 GB memory) and on a 64-bit Windows 10 computer

**Table 1.** Accuracy of algorithms.

Percentage of frames in which a body part was detected automatically. N indicates the number of frames of the string-pulling epoch that was analyzed. (N/A indicates not applicable and values were not used for calculating the overall accuracy. For these situations, either color of a body part was not separable in masks or due to limitation of the camera, hands were blurry in most of the frames).

| Mouse/Body Part | Body | Ears | Nose | Hands |
|---|---|---|---|---|
| Black Mouse 1 (N = 477 of 630) | 100 | 100 | 98.32 | 98.74 |
| Black Mouse 2 (N = 132 of 245) | 100 | 100 | 96.12 | 87.12 |
| Black Mouse 3 (N = 351 of 1435) | 100 | 97.14 | 92.88 | 81.48 |
| Black Mouse 4 (N = 191 of 629) | 100 | 97.38 | 100 | 89.53 |
| Black Mouse 5 (N = 171 of 808) | 100 | 100 | 98.83 | 93.57 |
| Simulated Black Mouse (N = 300 of 477) | 100 | 100 | 100 | 99.67 |
| *Mean (Black Mice)* | *100* | *99.08* | *97.17* | *91.70* |
| White Mouse 1 (N = 275 of 990) | 100 | 96.36 | 82.91 | 94.91 |
| White Mouse 2 (N = 300 of 594) | 99.67 | 99 | 100 | 91 |
| White Mouse 3 (N = 251 of 453) | 100 | 94.02 | N/A | 60.16 (N/A) |
| White Mouse 4 (N = 166 of 421) | 99.40 | 90.96 | N/A | 30.12 (N/A) |
| White Mouse 5 (N = 216 of 930) | 100 | N/A | N/A | 0 (N/A) |
| *Mean (White Mice)* | *99.80* | *95.10* | *91.45* | *92.95* |

with two processors (Intel Xeon CPU E5-2620 v4 @ 2.10 Gz) and 256 GB RAM. Training iterations were 1030000 and the train and test errors were as follows in pixels. For network trained to identify; 1) only hands, errors were 3.3 and 3.94, 2) only nose, errors were 4.19 and 8.74, 3) only ears, errors were 4.76 and 11.85, and 4) all objects, errors were 5.07 and 10.46.

For the epoch shown in previous figures for the Black mouse, the networks (separate for ears, nose, and hands) faithfully identified the ears, nose, and hands, although with some errors (*Figure 12*). Errors were mostly negligible for the identified position of ears and nose (*Figure 12B,C and F*). However, errors were up to ~100 pixels (pix) (10 mm) for the identification of hands in frames where the speed was large (*Figure 12A,D and E*). These errors can be reduced if further training of previously trained networks is done with more frames. For now, we offer an option to correct for these errors using the provided graphical user interface in which Deeplabcut results can be loaded and manually curated. Since our procedure (and software) for tagging requires manual validation of results, it can be used to provide ground truth data for validating the results of neural networks-based tagging of body parts.

## Graphical user interface of toolbox and procedure for processing video data

The details of the graphical user interface (GUI, *Figure 13*) of the toolbox and procedure for processing video data are presented in the online wiki of GitHub. The software can be downloaded from the following GitHub link along with two sample videos from Black and White mice whose analysis is presented in this paper (*Video 1* and *Video 2*). https://github.com/samsoon-inayat/string_pulling_mouse_matlab (*Inayat et al., 2019*; *Inayat, 2020a*; copy archived at https://github.com/elifescien-ces-publications/string_pulling_mouse_matlab).

The tagged videos for the Black and White mice are shown above (*Video 1* and *Video 2* respectively) and Youtube links are provided on the online wiki page. https://github.com/samsoon-inayat/string_pulling_mouse_matlab/wiki.

## General applicability of the software (whole-body and kinematic analysis)

To demonstrate the general applicability of the whole-body analysis and heuristic algorithms for tracking body parts, two videos not related to string-pulling were analyzed. In the first video, a

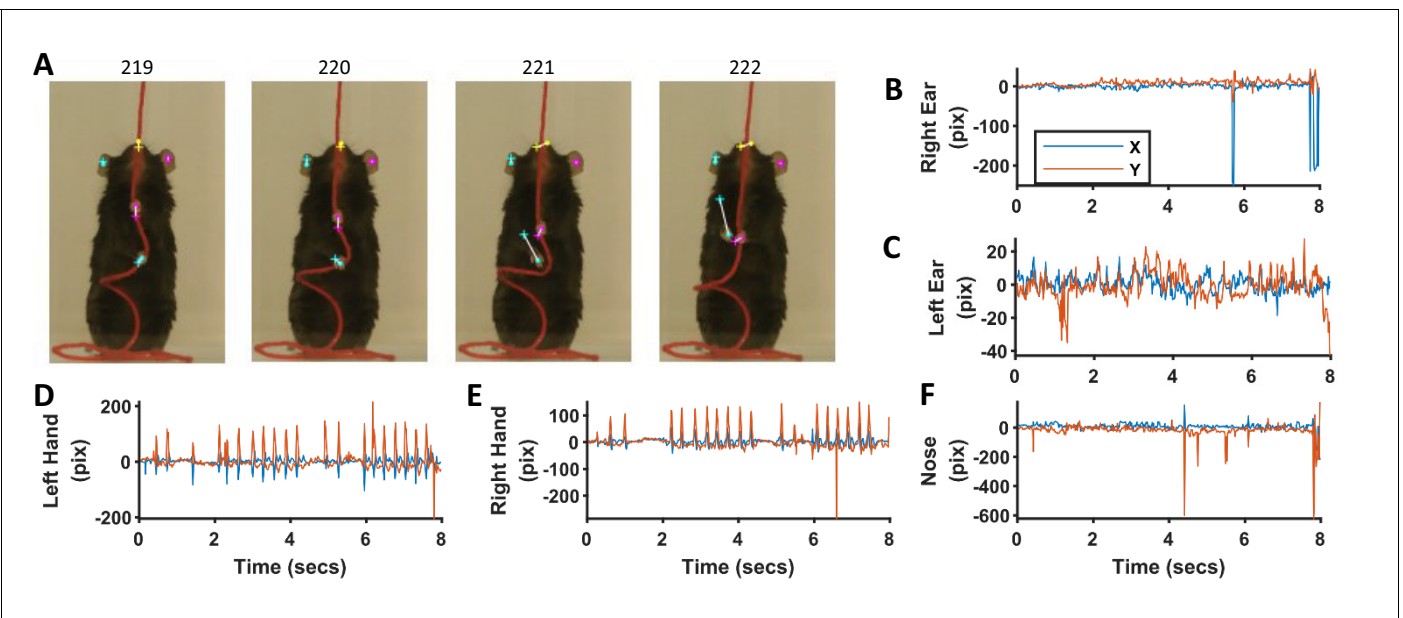

**Figure 12.** Comparison of identification of ears, nose, and hands with heuristic algorithms and machine-learning-based neural networks. (A) Representative frames highlighting the differences in position of ears, nose, and hands identified with the two methods. Dots for heuristics and plus for neural networks. (B–F) Differences in x and y coordinates of ears, nose, and hands identified with the two methods.

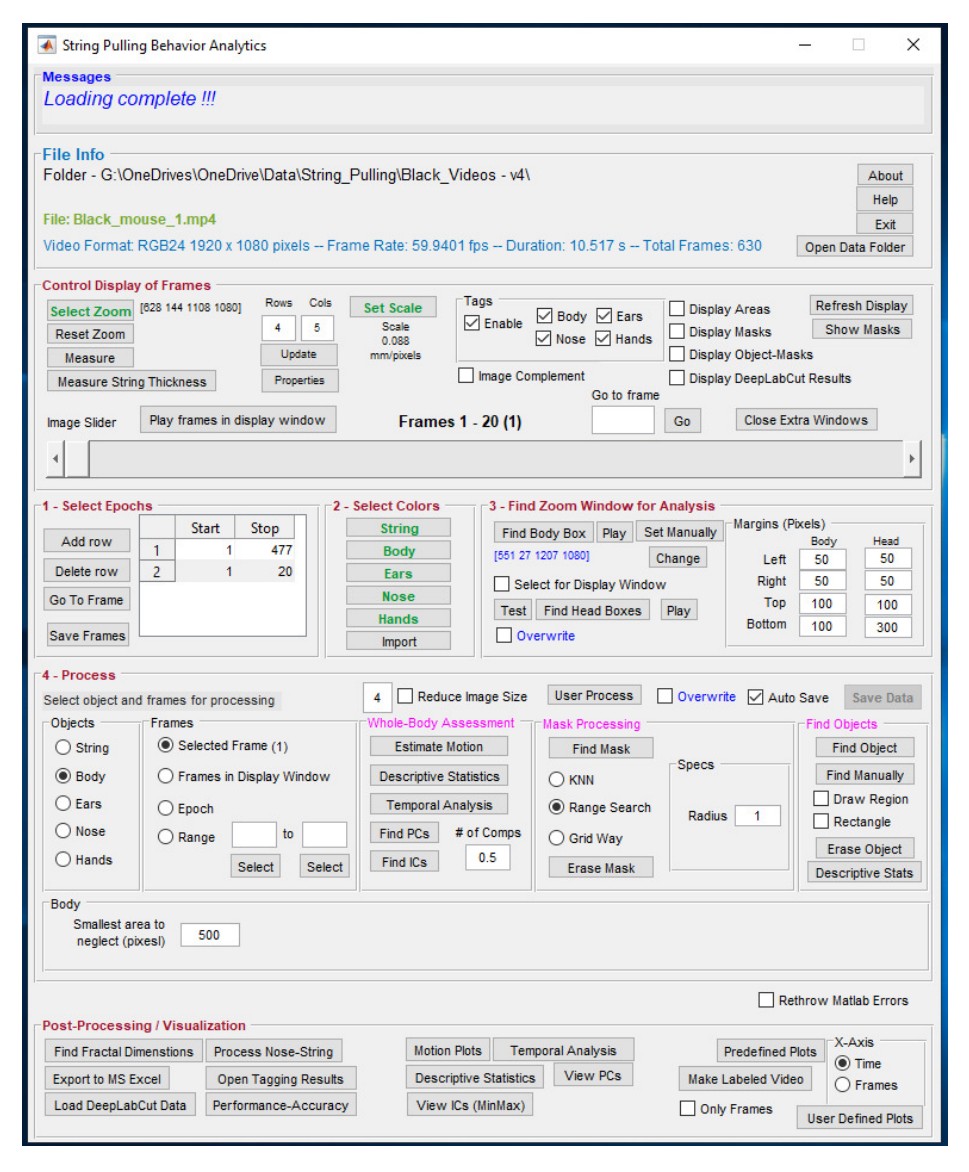

**Figure 13.** Graphical user interface of the toolbox. Individual steps for processing data are separated into panel blocks. The video frames can be viewed in the display window using the 'Control Display of Frames' panel. Step 0, the user has to define a zoom window for digital magnification of the video and scale for finding pixels/mm. Step 1, the user specifies epochs in which the mouse pulls the string. Step 2, the user defines colors. Step 3, the user finds zoom window, a sub region of the frames for which masks are calculated. Step 4, the user processes the epoch for whole-body analysis through 'Whole-Body Assessment' panel and also finds masks and later objects of interest. Finally, the user can generate plots using the 'Post-Processing' panel.

human in a sitting posture is catching and throwing a ball where the face, hands, and ball were tracked with 100, 99, and 95 percent accuracy respectively (*Figure 14*). In the second video, a mouse is tail clasped to freely hang and observed for the motion of hind paws. Here, body and hind paws were tracked with 100% and 88% accuracy, respectively (*Figure 14—figure supplement 1*). Links to both videos tagged with identified objects can be found on the following GitHub Wiki page. https://github.com/samsoon-inayat/string_pulling_mouse_matlab/wiki.

Processed data sets including whole-body analysis for both above-mentioned videos can be downloaded from the online folder 'General Applicability of Software' on OSF website (please see link above).

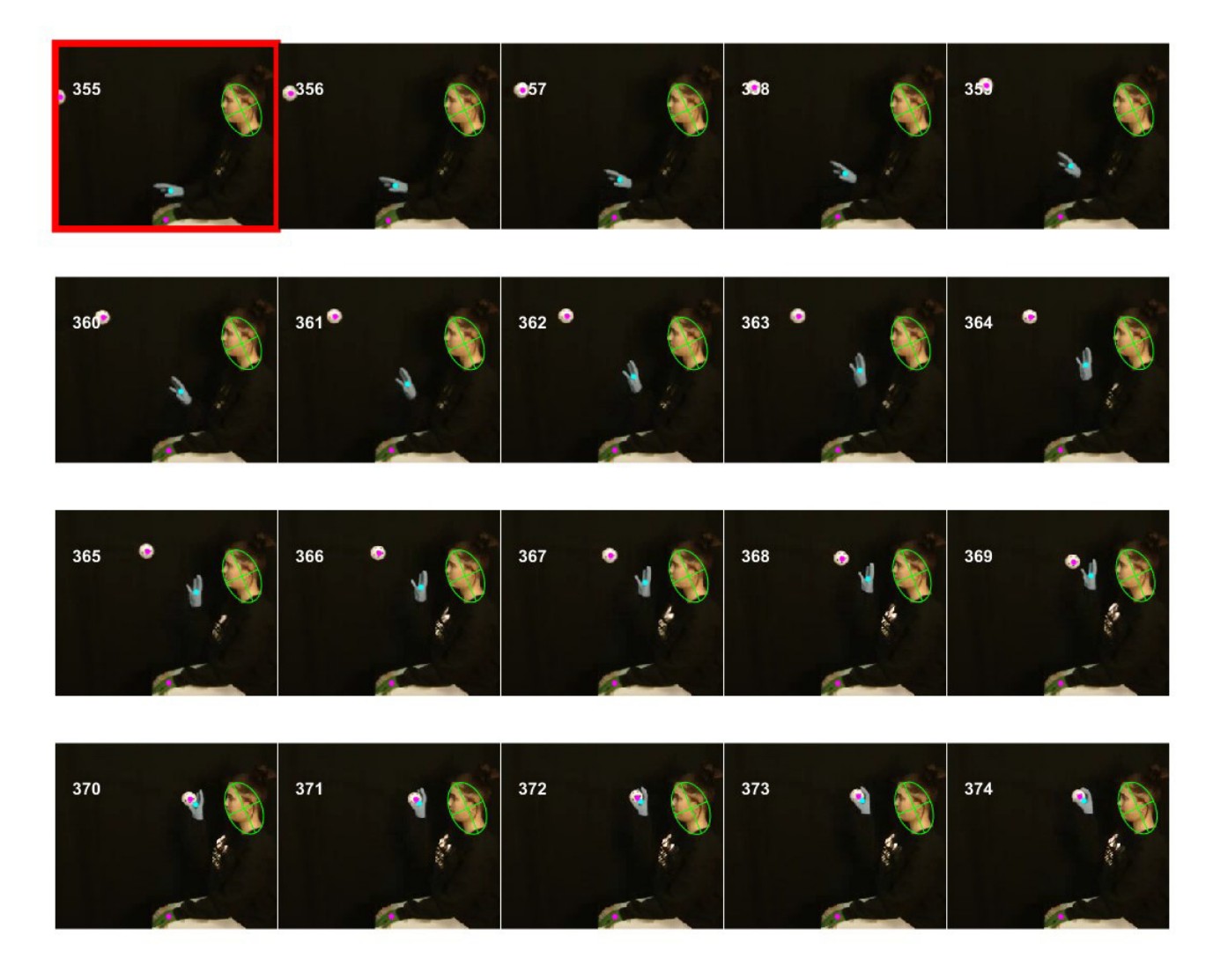

**Figure 14.** General applicability of the software for tracking objects shown in a video of a human catching and throwing a ball.
The online version of this article includes the following figure supplement(s) for figure 14:

**Figure supplement 1.** General applicability of the software for tracking objects shown in a video of a tail-clasped mouse where hind paws and body are tracked.

## Discussion

We present a Matlab based toolbox for seamlessly processing mouse string-pulling video data to perform whole-body assessment of position and speed as well as automatically identify the body, ears, nose, and hands in individual video frames. Quantification of the body and head postures is presented using estimates of body length and tilt, and head yaw, roll, and pitch angles while kinematic measures are determined for the hands. Because mice track the string with snout sinus hairs, changes in head posture relative to string motion provides an estimate of sensorimotor integration. Kinematic measures for body and head can also be calculated from the quantifications provided. This toolbox is provided under the GNU General Public Licence v3.0, and is open source, free, and any modifications are allowed as long as this paper is cited when publishing or distributing a modified version. Here, we discuss phenotypic differences in behavior in exemplar mouse strains used for

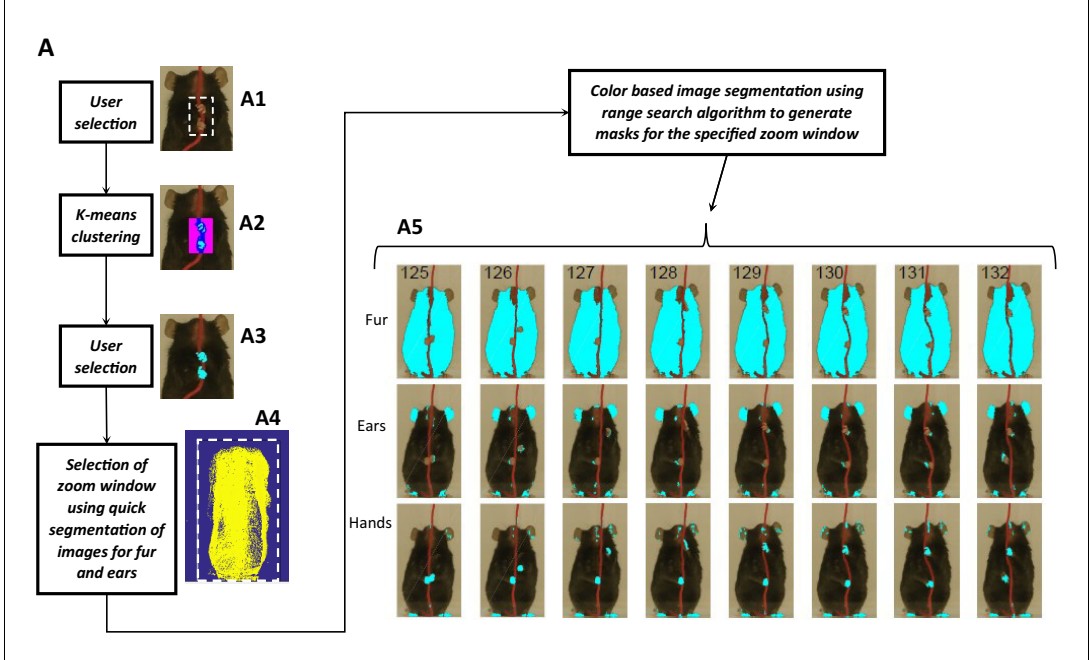

**Figure 15.** Image segmentation based on object colors for finding object masks and gross assessment from descriptive statistics of masks. (**A**) Color definitions in original frames are semi-automatically made with the user's assistance for fur, ears, nose, hands, and string. The user first selects a rectangular region around the object of interest (**A1**). K-means clustering is then applied to segregate different colors within the selected region (**A2**). The user then selects a color that corresponds to the object of interest (A3, example shown for selecting color for the hands). Using color selections for the fur and ears, quick segmentation is used to identify the fur and ears in every frame in an epoch of interest and projected onto a single frame after edge detection (**A4**). A rectangular window around the blob is drawn to define a sub region of frames for which masks are then found (A5, representative masks). The determination of rectangular window can also be done automatically and modified later if required. Note that masks are not always clean and include regions other than objects of interest for example, hand masks include regions identified in ears and feet because of color similarity.

The online version of this article includes the following figure supplement(s) for figure 15:

**Figure supplement 1.** Masks found with KNN, rangesearch, and custrom (grid-way) algorithm.

**Figure supplement 2.** Robustness of image segmentation shown in a video with different camera angle with a mouse on a table with brown base.

the analysis, characteristics and limitations of the toolbox as well as comparison with other similar software.

The description of behavior is central to understanding neural organization and function and accordingly behavior is frequently fractionated to describe the contributions of different body parts. The description of motor abnormalities emphasizes the allied contributions in which one body part compensates for impairment of other body parts (*Alaverdashvili and Whishaw, 2013*; *Jones, 2017*). For example, in using a hand to reach, detailed descriptions of hand movement also require a description of the contribution of postural adjustments of the body. Optimal string pulling can be envisioned as an act featuring individual arm/hand movements made with respect to a stable base of support featuring little body movement. Any variation of this relationship between the hands and body may be an expression of compensatory adjustment. The present procedure provides an integrated view of a prototool use behavior in which skilled hand movements are described along with the contributions of body movement, posture, and orientation.

We demonstrate phenotyping differences in the motor behavior of C57BL/6 (Back) and Swiss Webster (White) mice and present findings that White mice have larger variation in positions of the body, larger body speed but smaller reach-cycle time of string-pulling motion as compared to Black mice. The interpretation of this result is that the White mice are displaying an impairment in arm/hand movements relative to Black mice and so use upper arm and body motion for compensation. It is recognized that albino mice display anatomical abnormalities in the visual system (*Lee et al., 2019*), other physiological processes for example fatty acid uptake and trafficking (*Seeger and*

*Murphy, 2016*), and recovery after spinal cord injury (*Noristani et al., 2018*). There is also evidence that albino strains of laboratory animals display motor system abnormalities (*VandenBerg et al., 2002*; *Whishaw et al., 2003*). Thus, it is possible to speculate that abnormalities in motor systems, such as the corticospinal tracts that control the arms and hands, lead to compensatory whole-body movements of string pulling in the White mice, as is reported after motor cortex stroke in mice (*Farr and Whishaw, 2002*).

That the string-pulling procedure captures strain differences in the two exemplar strains of mice that are examined, in which Swiss Wister mice display more body motion relative to C57BL/6 mice, shows that the procedures can be used for phenotyping strain differences more generally. In addition, the behavior of string-pulling is homologous to string-pulling described in many other animal species including humans, suggesting that our method can be exploited for species comparisons and for developing other animal models of human neurological conditions (*Ryait et al., 2019*; *Singh et al., 2019*).

## The whole-body analysis approach

The whole-body analysis approach developed and presented here provides a generic method for analyzing animal behavior. With this approach, rather than analyzing kinematics of body movements, overall characteristics of the image sequence and derived spatiotemporal signals such as speed frames are assessed within a spatially confined region representing the average position/location of the animal. Thus, the need for pose estimation or tracking of body parts is eliminated. Intuitively, spatiotemporal signals with two spatial dimensions and a temporal dimension can be assessed with spatial and temporal measures and could also be examined using factorization to decompose into simpler components using for example principal and independent component analysis. To obtain a global and spatial summary over time, calculations were first performed in the temporal dimension to obtain single 2D frames for example with temporal measures of central and variability descriptive statistics, Fano factor, entropy, and Higuchi fractal dimension, for quantifying the regularity and variation, dispersion, randomness/disorder, and complexity respectively of the time series of pixel values. After these quantifications, a consistent method is required to compare them across animals and groups of animals.

The spatial mean values of the above-mentioned parameters over the average position or location of the animal is a reasonable overall spatiotemporal estimation, and as demonstrated above, allowed highlighting statistically significant differences between White and Black mice. Further differences between White and Black mice were evident with spatial measures of entropy, sharpness, and Hausdorff fractal dimension applied on previously obtained single 2D frames for quantifying their randomness/disorder, edges/blurriness, and complexity, respectively. One can envision mouse string pulling as a modification of climbing a string and accordingly, the more erect posture of the White mice uncovers a different strategy relative to strategy of the Black mice. Note that an expert eye might observe speed differences between Black and White mice from raw videos, but the spatiotemporal measurements demonstrated here provide statistical evidence not only for speed differences but also for gauging randomness and complexity of signals. Some of the measurements shown here were not statistically different between Black and White mice. However, for a different set of videos or an increase in sample number the differences might become significant statistically. Therefore, all the measures are included in the toolbox for users to apply and explore in the data at hand.

## Robustness and limitations of image segmentation and automatic detection of body parts

The success of the heuristic approach to detect body parts automatically is crucially dependent upon the quality of masks generated with image segmentation. Therefore, four strategies have been devised to improve image segmentation. First, an interface is provided to select colors of objects where the user can interactively select a region of interest, select a candidate set of colors, and then inspect the mask for accuracy (see methods). This way a narrow range of colors can be selected to represent an object. Second, a 'zoom window' is used to select a region of frame which encompasses the whole behavioral epoch of interest and therefore allows for fast and improved image segmentation (only look for colors where the user knows they are). An automatic method for detecting zoom window has been implemented where in each frame, the boundaries of the animal are found

based on its body set of colors (defined previously). The zoom window found automatically can be inspected and modified as well as can also be defined manually all together. Third, similar to the zoom window, a 'head box' is found to define a window containing only the head. Head box is also found automatically, can be later inspected and modified. The use of head box improves the efficiency of detection of ears as the search region is restricted.

Finally, three different methods for image segmentation (*Figure 15*) have been provided (KNN, rangesearch, and a custom algorithm) to allow the user to choose what might work best for the data set at hand. KNN and rangesearch algorithms work similarly to find colors of interest that are closest to the colors of choice based on 'Euclidian distance'. However, KNN searches for the nearest neighbors while in rangesearch the user predefines the Euclidian distance. For videos with excellent contrast that is clear separation of colors of body parts from background such as for the illustrative Black mouse shown here (see figures above), rangesearch with a Euclidian distance of 1.5 (default in GUI) works very well. However, KNN works better for frames where more rigorous search of closest colors is required. For poor videos where background has colors like those of body parts, the custom algorithm works best. This algorithm divides the screen using a grid and then uses rangesearch to find colors within each grid cell. Using a manually set threshold, this method eliminates spurious pixels and one gets sparser segments depending upon the size of the grid. Sparser segments allow for more accurate detection of islands that represent a certain body part. An illustrative example of finding masks with three methods is presented in *Figure 15—figure supplement 1* for the tail-clasped mouse to demonstrate the sparsity of segments found with the custom method. An example of finding all the body parts in a video collected with different illumination conditions, camera angle, and brown-colored table-top is presented in *Figure 15—figure supplement 2* which illustrates the effectiveness of the above-mentioned strategies of image segmentation.

Automatic identification of body parts features a heuristics approach which requires consistent input data, that is the videos be recorded in a consistent manner. The number of pixels in a frame, frame rate, illumination conditions, and overall position of the mouse in video frames, if consistent from video to video, will make processing effective. If these parameters are changed, image segmentation options (see above) and thresholds that are set in the software can be readjusted for improving accuracy. These thresholds are mentioned in the 'selectAppropriateRegions' function in the toolbox and in individual functions to identify body parts.

Due to the use of a heuristic approach, sometimes the algorithm fails to correctly identify regions for hands. In cases where a Matlab error occurs while identifying the hands, the user is given the option for manual tagging. Nevertheless, in case of no Matlab error, the algorithm can still incorrectly label a region as a nose rather than a hand. In such a case, the user has to manually observe frames for accurate detection and in case of erroneous tagging, use the manual tagging option provided in the GUI. Furthermore, the erroneous tagging in one frame can continue into subsequent frames because identification of the hands in a frame depends on correct identification in a previous frame. Therefore, the recommendation is to process the 20 frames that are displayed along with GUI together. In the sample data of the Black mouse, less than 2% of frames (6/477) were incorrectly labeled for hands and had to be tagged manually. The other case in which erroneous tagging might happen includes videos in which a mouse is pulling a string when turned sideways. In this case only one ear is visible. Then, when both ears appear again, the algorithm still detects one ear. The user has to again manually initialize the detection of the other non-detected ear for the software to automatically detect the ear in subsequent frames.

## Video acquisition strategy to improve accuracy of automatic detection of body parts

Since, image segmentation and the heuristic algorithms rely on high-contrast image sequence, the choice of colors for background and string can affect the outcome for example if for a White mouse, white background or string is chosen, the software would not be able to distinguish between string and body (fur of the animal). Similarly, if shutter speed of the camera is not high enough, frames in the image sequence will have blurry objects particularly hands as they have high-speed motion. The software will then have difficulty in resolving colors of hands from blurry images.

Illumination conditions should also be such as to enhance the contrast of acquired images. We recommend using front facing cold light (on the same side as the camera), that is light is in front of the animal. High resolution of the video is also preferred for better pixel resolution for resolving

body parts (HD 1920 x 1080 is recommended). Note that these requirements are important for identifying and tracking body parts in videos but whole-body assessment can still be done on low- to medium-quality videos.

## Comparison with artificially intelligent machine learning based software

Machine learning (ML) based software for pose estimation in videos such as 'DeepLabCut' (*Mathis et al., 2018*) and 'Animal Part Tracker' (*Kabra et al., 2013*) and for object recognition such as 'DeepBehavior' (*Arac et al., 2019*) can provide a seamless work flow, but they are computationally intensive initially when training is required. For identifying a new/different object, or for improving the accuracy of detection (*Figure 12*), usually retraining must be done. The algorithms presented here are simpler, require smaller computations, and can be implemented on any computer with reasonable computing power and temporary storage for example 2 GHz processor and 32 GB of RAM. In comparison, ML-based software require high-end computers ideally with graphical processing units.

In contrast with DeepLabCut and AnimalPartTracker, our approach for characterizing string-pulling behavior has several advantages. Instead of just pose estimation, it provides direct measurements of relevant parameters that might be used to identify similarities and differences between control and experimental groups of animals for example whole-body analysis, use of optical flow to identify speeds, identification of body length and angle, head posture, and kinematic analysis of body and hand movements. To identify body length with ML based software, one would have to label many points around the body and track them in frames to identify the body boundary. In short, the method imposes a priori an interpretation of the behavior, whereas our segmentation methods is neutral and so is more objective in this respect.

With body color definition and image segmentation, identification of the body is a breeze, and features almost 100% accuracy. For other body parts as well, ML-based pose estimation would track points, whereas with image segmentation one can track regions/areas which with the former method must be done in a secondary analysis. The approach presented here also has a faster lead time. The user can obtain results the same day after data acquisition as our software does not need any prior training of a neural network (which can take several days), rather one can open videos and start analyzing right away. Depending upon the quality of videos, 2–4 hr might be required to analyze 500 frames. Although, after training a neural network, individual videos might take smaller amount of time for pose estimation, post manual validation and subsequent analysis must be done for characterizing behavior. In the provided toolbox, pdfs of final high-quality figures are made and stored for immediate use and scripts are provided to compare groups of animals. The faster lead time with the current approach would also apply were experimental conditions changed for example animal species or strain, whereas retraining of the neural network would be required.

## Data compression for efficient storage of masks

While writing this software, data compression techniques were used to minimize storage of calculated masks in addition to the provision provided to select a zoom window which uses a subset of frames to process. Since each pixel of a mask is either 1 or a 0, it can occupy a single bit in a 1-byte number. This way individual bits of a byte represent a single pixel of an individual mask while the whole byte can be used to store one pixel for eight masks. Using this methodology, five masks are stored in an array with each element one byte long.

## Support and software maintenance, future work, and possible extensions of the toolbox and the string-pulling behavioral task

A user manual of the software has been provided in the online wiki at the GitHub page along with a video tutorial. Support for issues with the software will be provided via email or through the GitHub support page where other people will be able to view issues and their solutions. The software will be maintained as per need and any further improvements and changes will be updated on the GitHub 'readme' and 'revisions' page. In case major revisions are made to the software regarding GUI or improvements in detection algorithms, a completely newer version will be provided on GitHub and the users will be redirected to the newer GitHub page. An undisclosed email list serv of

the users will be kept by the corresponding authors. Users will be encouraged to contribute toward future development and will be added as contributors to the GitHub page.

Future development regarding whole-body analysis might include assessing spatiotemporal properties of the phase of velocity vectors which would add directional information to motion measurements. For the tracking of body parts, additional functionality could be added regarding detection of phases of string-pulling movements for example when the hands grasp and release the string. String masks are available in the toolbox to add this functionality. The heuristic algorithms can also be easily modified by changing the appropriate functions. The string-pulling task can also be enriched with sensory, motor, and behavioral complexity. For example, sensory complexity would involve strings with different cross-sections, texture, or coated with odors. Motor complexity would involve adding tension or variable load to the string. Behavioral complexity would involve a paradigm in which the animal has to learn to associate string-pulling with a reward for example water.

Social competition or social altruism can also be studied with string-pulling. For example, in a competitive strategy, two animals will pull the string from two ends to obtain reward attached in the middle. The scenario could be presented to study motivation before and after stress. In experiments to study altruism, string pulling of one animal could give the reward to another animal. Motor memory formation and the role of sleep in its consolidation could also be studied (*Inayat et al., 2020b*). We anticipate that this toolbox holds promise for facilitating high-throughput platform for the behavioral screening of drugs that may help in the treatment of Parkinson, Huntington, Alzheimer and similar diseases in addition to brain injuries (i.e. ischemic stroke, spinal cord injury, etc.). In short, the method can reveal the extent to which the hands can be used independently of the body, the asymmetry of hand use, the sensory control of reach guidance, the speed/accuracy of hand use, and compensatory adjustments between the individual measures. Exemplar assessments are relevant to estimating the effects of brain injury and disease, recovery and treatment effects, as well as strain and species differences in string pulling behavior.

# Materials and methods

## Experimental animals
Adult mice 3–6 months old, five male C57 Bl/6J (Black) and five male Swiss Webster Albino (White), were used in this study. No explicit power analysis was used to estimate sample size because in this work our goal was to establish analysis methods. N = 5 was chosen for both strains of mice due to limited availability of White mice. We observed a statistically significant effect of genotype for many parameters (see results) with the chosen sample size. To examine magnitude of effect where statistical significance was observed, we also report effect size.

Animals were kept in a controlled temperature (22°C), humidity, and light with a 12:12 light/dark cycle. They were singly housed as they were food restricted to increase the yield of string-pulling behavior. All testing and training were performed during the light phase of the cycle. All experiments were performed in accordance with the Canadian Council of Animal Care and were approved by the University of Lethbridge Animal Welfare Committee.

## String-pulling task
The apparatus and training procedure were as described previously (*Blackwell et al., 2018a*). Briefly, the test box consisted of a clear Plexiglas cage (20 cm long, 9 cm wide, and 20 cm high) in which a string was thrown over the center of the front wall so that its end could be reached by the mouse. A video camera was placed in front of the cage and behavior was filmed with a Panasonic camcorder at 60 f/s with a 1/1000 shutter speed and frame resolution of 1920 × 1080 pixels. For three White mice, the data was collected with another camera, a Sony Camcorder with 1/750 shutter speed and 60 f/s. Mice would spontaneously pull the string into the cage but were also lightly food restricted so that they were motivated to pull the string to obtain a food reward (pieces of a cashew nut or cheerios) attached to the end of the string.

## Analysis platform and preliminary processing
Matlab R2016 and Microsoft Windows platform was used for the development of the toolbox and analysis of the video record. Later, it was tested and upgraded to work in Matlab R2018b and

Matlab 2019b. From the videos, a subset of frames representing string-pulling epoch was selected. The start of the epoch was when the mouse touched the string and started pulling it down while the end of epoch was when the mouse stopped the bout of string-pulling. Furthermore, a subregion of frames was selected for analysis to reduce computation time. In the toolbox, this subregion is called 'auto zoom window' and is found automatically but is also subject to adjustment.

In order to do a fair comparison between the performances of Black and White mice with whole-body analysis, the image frames of Black mice were complemented using Matlab's 'imcomplement' function. To accelerate the processing of whole-body analyses, red, green, and blue (RGB) frames were first resized to one-fourth their original size using Matlab's 'imresize' function and then converted to gray scale frames using Matlab's 'rgb2gray' function. The user can, however, set the image resize value using the provided graphical user interface (see below). For kinematic assessment of movements, body, ears, nose, and hands were tracked in the original frames using color-based image segmentation and heuristic algorithms.

## Optical flow analysis of image sequence for estimation of velocity vector fields and speed frames

The Combined Local-Global (CLG) method was used to estimate optical flow in string pulling videos (*Bruhn et al., 2005*; *Afrashteh et al., 2017*). The Matlab implementation of the CLG method provided by *Liu, 2009* was used with the following default parameters which the user can modify if required (alpha = 0.01, ratio = 0.5, minWidth = 20, nOuterFPIterations = 7, nInnerFPIterations = 1, nSORIterations = 30). This method was chosen because it provides a better optical flow estimate compared to Horn-Schunck and Temporospatial methods (*Afrashteh et al., 2017*). The chosen parameter values faithfully estimated optical flow, as was validated with visual observation of moving objects in frames. To reduce computational time, image frames in videos were reduced by four times but resizing of image frames did not affect the quality of images or the calculations of optical flow as confirmed with visual observation of velocity vector fields. As the optical flow computes one velocity vector field frame from two video frames, the number of optical flow frames that is velocity vector fields, is one less than that of video frames.

## Descriptive statistics

The mean, median, mode, standard deviation, skewness, and kurtosis were calculated for image sequence and speed frames by using respective Matlab functions. For example, the mean frame was determined by finding the mean of the time series of all individual pixels in a frame sequence that is $\mu_{x,y} = \frac{1}{N}\sum_{t=1}^{N} I(x,y,t)$ where $\mu_{x,y}$ denotes mean value of intensity at x and y location, t is frame number, N is the total number of frames, and I is the average gray scale intensity determined from red, green, and blue frame sequences. Other parameters were similarly calculated.

### Finding a mask for mean mouse position from mean frame

A mask was generated to estimate the average position of a mouse by using a threshold equal to average of all pixels in the mean frame. Matlab's 'imbinarize' function was used to generate the mask, which makes all pixels zero whose value is below the threshold and sets other pixels to a value of one.

## Temporal analysis of image sequence with Fano factor, entropy, and Higuchi fractal dimension

To assess the temporal changes in the position and speed from image sequence and speed frames respectively, three measures, Fano factor, entropy, and Higuchi fractal dimension (HiFD) were used. A Fano factor frame for the image sequence was calculated to find the Fano factor value for each pixel from its time series using the following equation; $Fano\ Factor = \frac{\sigma^2}{\mu}$, where $\sigma$ is the standard deviation and $\mu$ is the mean of time series. Similarly, an entropy frame for the image sequence was calculated by finding the entropy of the intensity time series of individual pixels by using the 'entropy' function of Matlab which uses the following relationship; $E_{x,y} = -\sum p_{x,y} log2(p_{x,y})$ where $E_{x,y}$ is the entropy of the time series at x,y location and $p_{x,y}$ is the normalized histogram counts of intensity values for the same location returned by the 'imhist' Matlab function. Prior to finding this

measure, 'mat2gray' Matlab function was applied on time series to scale maximum and minimum values as 1 and 0, respectively. The Higuchi fractal dimension (*Higuchi, 1988*) frame was also calculated from the time series of individual pixels in a similar way using a function downloaded from https://www.mathworks.com/matlabcentral/fileexchange/50290-higuchi-and-katz-fractal-dimension-measures. The value of Kmax parameter used to find HiFD was arbitrarily chosen to be 10 (*Spasić et al., 2005*). The measures of Fano factor, entropy, and HiFD were also calculated in the same way for speed frames.

To statistically compare the values of the above-mentioned parameters between groups of Black and White mice, values of parameters were extracted from respective frames using the mean position mask (see above). For example, for comparing entropy, values of entropy were selected from the entropy frame from pixels that had a corresponding value of 1 in the mean position mask. The mean of all these entropy values was chosen as a representative value for the animal. These means were statistically compared between groups using Student's t-test.

## Principal component analysis of image sequence and speed frames

In Matlab, the 'pca' function was used to determine principal components (PCs) of the image sequence and motion profile (speed frames) determined with optical flow analysis. For a frame sequence, the 3D matrix HxWxN (H, W, and N being height, width, and number of frames) was converted to a 2D matrix with the number of rows equal to HxW and columns equal to N. The function pca was then executed to determine N principal components. The individual component scores were then reshaped from 1D (HxW) to 2D H rows and W columns. PCs of speed frames were similarly calculated.

## Independent component analysis of image sequence and speed frames

A fast independent component analysis (fastICA) algorithm (*Hyvärinen and Oja, 2000*) was used to find independent components (ICs) of the image sequence as well as speed frames. The function was downloaded from https://www.mathworks.com/matlabcentral/fileexchange/38300-pca-and-ica-package. To apply 'fastICA' on the image sequence or speed frames, the 3D matrix of frames was converted to a 2D matrix with the number of rows equal to HxW and columns equal to N where H and W are the number of height and width pixels while N is the number of frames. Here N was equal to 50% of the original number of frames in the image sequence or speed frames that is number of independent components was chosen to be 50% of the number of frames in the input sequence. This was done to reduce computational time. After applying the fastICA function, individual component frames were determined by reshaping from 1D scores (HxW) to H rows and W columns. Since, the algorithm to find ICs is iterative, the minimum tolerance set to stop iterations was 5e-5 and the maximum number of iterations was set to 1000. ICs were found before reaching the maximum number of iterations that is the error went below tolerance.

## Calculation of spatial measures of entropy, sharpness, and Hausdorff fractal dimension

For various frames obtained in the previous sets of whole-body analyses; for example mean frame of image sequence or the first principal component frame, spatial entropy, sharpness, and Hausdorff fractal dimension (HFD) were calculated. The same entropy function described earlier was used for finding spatial entropy after scaling the frame values between 0 and 1 with 'mat2gray' Matlab function. Sharpness was calculated using the following equation:

$$S_n = \left[ \sum_{x,y} \sqrt{|\nabla I_{x,y}|} \right] / (H \times W),$$

where $\nabla I_{x,y}$ represents gradient of intensity image and H and W denote height and width of the image respectively. Function to calculate sharpness was downloaded from https://www.mathworks.com/matlabcentral/fileexchange/32397-sharpness-estimation-from-image-gradients. The Hausdorff fractal dimension was obtained using the 'boxCountfracDim' function downloaded from https://www.mathworks.com/matlabcentral/fileexchange/58148-hausdorff-box-counting-fractal-dimension-with-multi-resolution-calculation. This fractal dimension uses the following relationship; HFD = -log (N(ε))/log(ε) where N is the number of pixels above zeros in a box and ε is the box size. In the box

counting method, an image is first binarized using an appropriate threshold and then non-overlapping boxes of different sizes (different ε values) are overlayed on the image and the number of pixels above zero value are counted to obtain N(ε). A line is then fitted for a scatter plot of log(N(ε)) and log(ε) and the slope of that line provides the fractal dimension (FD). Here, the threshold used to binarize an image was the mean value of all its pixels. For the Black mice, the frames were complemented before the calculation of HFD for comparison with White mice.

## Synthetic data for testing and validation of algorithms

To test the efficacy of the software, two different types of data sets were generated, 1) empirical: based on experimental values of mouse body position and those of hands, nose, and ears, and 2) modelled: where the movement of hands were calculated by a sinusoidal function. For both data sets, color of fur/body, ears, nose, and hand were chosen from the range of observed values to create elliptical patches of body parts on a rectangular background patch. For empirical data, animal movements were created using the centroid, major- and minor- axis of the ellipse calculated from the real experiments. For modeled data, we observed that x- and y- motion of hands can be modelled using two 180° out of phase sinusoidal oscillators, and body tilt is in-phase with the reaching-hand. In general, we observed that one complete reach-grasp-release cycle was completed in ~12 frames (0.2 s). Thus, the 360° x- and y- motion was divided into 12 parts to mimic the hand movements. Further, hand movements were linked to the body centre to mimic the changes in hand movements associated with body movements.

## Color-based image segmentation for gross motion assessment and tracking of body, ears, nose, and hands

In computer vision, image segmentation is a basic process in which an image is divided into regions based on distinguishing features, such as color and texture. In the current video data, we utilized color-based segmentation of frames for finding regions containing fur, ears, nose, hands, and the string. To define a color, a user first selects a rectangular region around the object of interest (*Figure 15*) for example, hands. K-means clustering is then used to separate colors in the rectangular region into three clusters. With trial and error, it was found that three clusters were sufficient to separate colors of interest for body and hands while seven clusters were sufficient for separating colors of interest for the string and nose. The number of clusters can be changed with the user interface, however. The user is then allowed to choose the cluster that represents the object of interest for example, cyan cluster in *Figure 15* A3 for hands. Using these colors, a mask is then generated and shown to the user for visual inspection. Through the supplied interface, the user can then add or subtract colors until a reasonable mask is generated. The user can then move to the next frame and repeat the process for the new frame. This way, with five to ten frames within an epoch chosen randomly, an appropriate selection of colors for an object of interest is made. All color values in a cluster are stored and later used to find masks. In a similar fashion, the user selects colors for other objects of interest.

Once colors are defined, masks are generated in which colors matching the color of interest are found (*Figure 15* A5). Three different methods can be used to generate masks; 1) K-Nearest Neighbor (Matlab's KNNsearch function), 2) a Range search algorithm (Matlab's rangesearch function), and 3) a custom algorithm (custom Matlab function find_mask_gridway) which divides a frame into a grid and uses rangesearch for individual grid blocks. To save computational time and memory storage, masks are not found for the whole frame, rather for a subset of each frame. The region of a frame for which masks are determined is found such that it includes the mouse body in all frames of an epoch. First, the range of motion of the whole mouse body within an epoch is estimated by finding the edges of a mouse's body in each frame of an epoch and plotting them together (*Figure 15* A4). The user then draws a window around the blob of plotted edges, which is then stored as a parameter in 'auto zoom window' and used to find masks. On a computer with Intel Core i7-4820K with two CPUs (each 3.7 GHz) and 32 GB RAM, it takes ~1 hr to process 477 frames (each frame being 457 × 895 pixels) for finding five sets of masks for fur, ears, nose, hands, and string. With an option to reduce image size by a factor adjustable with the user interface, the calculation of masks can be accelerated by several orders of magnitude.

## Statistical analysis

Statistical analysis was done using Matlab R2016b. Data are presented as mean ± the standard error of mean (Mean ± SEM). Two-sample Student's t-test was used for statistical comparisons and an alpha value of <0.05 was used to determine significance. Effect size was measured using Cohen's d and reported for substantive results.

## Acknowledgements

We thank Di Shao for animal husbandry and Behroo Mirzagha for her help in collecting video data.

## Additional information

### Funding

| Funder | Grant reference number | Author |
| --- | --- | --- |
| Canadian Institutes of Health Research | 390930 | Majid H Mohajerani |
| Natural Sciences and Engineering Research Council of Canada | 40352 | Majid H Mohajerani |
| Alberta Innovates | 43568 | Majid H Mohajerani |
| Alberta Alzheimer Research Program Grant | PAZ15010 | Majid H Mohajerani |
| Alberta Alzheimer Research Program Grant | PAZ17010 | Majid H Mohajerani |
| Alzheimer Society of Canada | 43674 | Majid H Mohajerani |

The funders had no role in study design, data collection and interpretation, or the decision to submit the work for publication.

### Author contributions

Samsoon Inayat, Conceptualization, Data curation, Formal analysis, Supervision, Validation, Investigation, Visualization, Methodology, Writing - original draft, Project administration, Writing - review and editing; Surjeet Singh, Conceptualization, Formal analysis, Validation, Investigation, Visualization, Methodology, Writing - review and editing; Arashk Ghasroddashti, Data curation, Formal analysis, Validation, Methodology, Writing - review and editing; Qandeel, Pramuka Egodage, Data curation, Investigation, Methodology; Ian Q Whishaw, Conceptualization, Resources, Formal analysis, Supervision, Validation, Investigation, Visualization, Methodology, Writing - review and editing; Majid H Mohajerani, Conceptualization, Resources, Supervision, Funding acquisition, Investigation, Project administration, Writing - review and editing

### Author ORCIDs

Samsoon Inayat (iD) https://orcid.org/0000-0003-1966-7967
Surjeet Singh (iD) https://orcid.org/0000-0002-4646-2089
Majid H Mohajerani (iD) https://orcid.org/0000-0003-0964-2977

### Ethics

Animal experimentation: All experiments were performed in strict accordance with the Canadian Council of Animal Care and were approved by the University of Lethbridge Animal Welfare Committee (Protocol 1812).

### Decision letter and Author response

Decision letter https://doi.org/10.7554/eLife.54540.sa1
Author response https://doi.org/10.7554/eLife.54540.sa2

## Additional files

### Supplementary files
• Transparent reporting form

### Data availability

The software is available to download from https://github.com/samsoon-inayat/string_pulling_mouse_matlab copy archived at https://github.com/elifesciences-publications/string_pulling_mouse_matlab. All video source and processed data is made available at the following website https://osf.io/gmk9y/.

The following dataset was generated:

| Author(s) | Year | Dataset title | Dataset URL | Database and Identifier |
|---|---|---|---|---|
| Inayat S | 2019 | Mouse String-Pulling Data | https://doi.org/10.17605/OSF.IO/GMK9Y | Open Science Framework, 10.17605/OSF.IO/GMK9Y |

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
