## [Decision Letter]

**Acceptance summary:**

String-pulling by rodents is a canonical task in the assessment of spinal cord function in normal and neurologically compromised mice. The laboratories of Whishaw and Mohajerani demonstrate algorithms to automate the study of postural and hand kinematics in string-pulling behavior. This adds a new dimension to the routines available for high-throughput, high-volume studies of behavior.

**Decision letter after peer review:**

Thank you for submitting your article "A Matlab-based toolbox for characterizing behavior of rodents engaged in string-pulling" for consideration by *eLife*. Your article has been reviewed by two expert and respected peer reviewers, and the evaluation has been overseen by David Kleinfeld as Reviewing Editor and Kate Wassum as the Senior Editor. The reviewers have opted to remain anonymous.

The reviewers have discussed the reviews with one another and with the Reviewing Editor, and they find considerable merit in your Resources contribution, which addresses quantification of a behavioral paradigm of increasing importance and use in the spinal cord circuit community. However, the reviewers raise a number of strong and legitimate concerns that we ask be addressed. The Reviewing Editor has drafted this decision to help you prepare a revised submission.

Coordination of limb movement is a hallmark of spinal cord motor control. Rope pulling, which involves such coordination in addition to proper posture and grasping, is a critical task in the study of spinal circuits. This work describes a set of software tools for the automated characterization of this behavior, and thus provides a valuable tool for the behavioral characterization of spinal circuits and experimental modification of those circuits.

Essential revisions:

Some of these steps will require new data analysis.

Please clarify the specifics of your tracking accuracy. "The paper states that "our methods can provide group truth data for validating the results of neural network-based tagging of body parts." It is not clear how this conclusion is derived."

Please address the robust nature of your approach. "It is unclear how robust the segmentation is to variations in the video, such as camera angle relative to the mouse, illumination conditions, and color of the mice. For example, the paws of the white mice could not be reliably tracked, and the authors had to use markers to paint the paws. This suggest variations in the video could strongly affect the outcome."

Please simplify the statistics and justify all measures. "… while central-tendency results show large difference in speed between Black and White mice, it is unclear how spatial entropy, sharpness, Hausdorff fractal dimension, Higuichi fractal dimension, Fano Factor, and Entropy support this effect". Nor is the necessity of bringing these statistical measures made clear.

Please bring trial-averaged data. As string-pulling is a highly repetitive near rhythmic behavior, it would be useful to see the average cycle of the behavior to understand the stereotypical hand movements. Such response could be computed from the data of Figure 10 and 11 by warping each pull cycle to a common time base. Then "compare descriptive statistics to quantify differences in hand movement trajectories across White and Black mouse groups."

Please discuss the generality of your software. It would be useful to see if the software for string pulling could be adapted to "… analyze limb movements during pasta handling, thereby showing general applicability of the software."

Please discuss the support and maintenance. Further, the poor documentation must be improved and the associated GitHub must be organized

Please explicitly spell out the advantages of your approach over related methods. In this regard, address the black-box method of "DeepLabCut" and the "Janelia Animal Part Tracker".

Lastly, please make the toolbox available to be downloaded in Matlab 2019b (reviewers were unable to successfully download).

[Editors' note: further revisions were suggested prior to acceptance, as described below.]

Thank you for resubmitting your work entitled "A Matlab-based toolbox for characterizing behavior of rodents engaged in string-pulling" for further consideration by *eLife*. Your revised article has been evaluated by Kate Wassum as the Senior Editor, and a Reviewing Editor.

The manuscript has been improved but there are some remaining issues that need to be addressed before acceptance, as outlined below:

The reviewers would like you to please provide more discussion of the significance of these findings for an average audience. The data show how whole-body measurements could be used to reveal differences in motor kinematics between mouse strains. These analyses suggest that Swiss Webster Albino (white) mice exhibit faster movements that are more localized to the forelimb compared to black 6 mice. What is the significance of these differences? For example, the authors interpret these differences as the white mice using a "more primitive strategy" than the black mice. In addition, the white mice were considered to have "exaggerated compensatory adjustment" and "impaired individualized arm/hand movements". Please elaborate on how these conclusions were derived and what are the implications for future studies?

We also noted a number of typos and errors in figure references. Here are some examples:

– In the first paragraph of the subsection “Comparison with artificially intelligent machine learning based software”, Figure 2.11 is likely referring to Figure 12.

– At the end of the first paragraph of the subsection “Robustness and limitations of image segmentation and automatic detection of body parts”, the figure is incorrectly referenced. It is likely referring to Figure 15—figure supplement 1.

–The figure on Time Warping should be Figure 11—figure supplement 1, not Figure 15.

---

## [Author Response]

Essential revisions:Some of these steps will require new data analysis.Please clarify the specifics of your tracking accuracy. "The paper states that "our methods can provide group truth data for validating the results of neural network-based tagging of body parts." It is not clear how this conclusion is derived."

We have updated the section “Benchmarking of algorithms for finding body parts” to clarify how tracking accuracy is calculated. We describe that with manual validation, frames are found where the algorithm “correctly” identified a body part and hence accuracy is the percentage of these frames. Furthermore, in the section “Neural networks for identifying ears, nose, and hands”, we describe that because manual validation is part of the procedure we have developed, our software can be used to generate “ground truth” data.

Please address the robust nature of your approach. "It is unclear how robust the segmentation is to variations in the video, such as camera angle relative to the mouse, illumination conditions, and color of the mice. For example, the paws of the white mice could not be reliably tracked, and the authors had to use markers to paint the paws. This suggest variations in the video could strongly affect the outcome."

We have added a paragraph discussing the robustness of the segmentation procedure in the updated Discussion section **“**Robustness and limitations of image segmentation and automatic detection of body parts”.

Mainly, we discuss how we improve image segmentation by 1) using an interface with which colors representing an object can be narrowly selected, 2) setting of a zoom window to select a region of frames containing behavioral epoch of interest, 3) setting a head box for individual frames containing head of the animal, and 4) three different methods for finding masks or regions of interest (illustrated in added Figure 15—figure supplement 1).

Regarding the coloring of paws of White mice, they were not detectable due to the presence of white hair on the dorsal side (almost fully covered thus low contrast) and not because of illumination conditions or camera angle. We also demonstrate robustness of image segmentation with mouse tail clasping video which we added to demonstrate the general applicability of the software (please see our fifth response below and Figure 14—figure supplement 1). Furthermore, in another test video with a camera angle such that the surface of a brown table was visible and shadows of mouse could be seen in the plexiglass box (please see Figure 15—figure supplement 2), we identified body, ears, nose, and hands with 100, 100, 100, and 95 percent accuracy in the 40 frames that we analyzed for testing.

It is important to note that with this program, investigators are not prisoners of filming constraints, targets can be colored as with the White mice to enhance filming contrasts.

Please simplify the statistics and justify all measures. "… while central-tendency results show large difference in speed between Black and White mice, it is unclear how spatial entropy, sharpness, Hausdorff fractal dimension, Higuichi fractal dimension, Fano Factor, and Entropy support this effect". Nor is the necessity of bringing these statistical measures made clear.

We thank the reviewers for this comment. We have updated sections 2.2, 2.3, and 2.4 where more interpretation of results has been provided e.g., how statistical differences between parameters highlight differences in dynamics of pixel intensities and speed signals depicting changes in positions and speeds of Black vs. White mice respectively. We have also added a new Discussion section “The whole-body Analysis Approach” where spatial and temporal measures that have been used are explained and justified. We argue that although in the current data set, some of the measures do not show a statistically significant difference between the two groups of mice, Black and White, in different data sets or with increased sample number, one might find significant differences. Therefore, for the sake of completion, we have included all the measures in the toolbox as part of the “whole-body analysis approach” which we highlight in Figure 1 summarizing analysis framework.

We also note that by using the excel data files, an investigator can explore any of a number of task relevant analyses including, independent movement of the hand relative to the body, asymmetry in hand movements, missed grasp vs. successful grasps, compensatory movements of the body relative to hand movement, sensory monitoring of the string using comparisons of snout movement vs. string movement, and so on.

Please bring trial-averaged data. As string-pulling is a highly repetitive near rhythmic behavior, it would be useful to see the average cycle of the behavior to understand the stereotypical hand movements. Such response could be computed from the data of Figures 10 and 11 by warping each pull cycle to a common time base. Then "compare descriptive statistics to quantify differences in hand movement trajectories across White and Black mouse groups."

Please see updated Figure 11 where trial averaged data is now shown and compared between Black and White mice. Although the mean amplitude of movement of hands is smaller for White compared to Black mice, it is not statistically significant. We have also performed dynamic time warping and included average cycles of Black and White mice in Figure 11—figure supplement 1. The amplitudes of Black and White mice were not statistically different.

Please discuss the generality of your software. It would be useful to see if the software for string pulling could be adapted to "… analyze limb movements during pasta handling, thereby showing general applicability of the software."

Please see new Results section “General applicability of the software (whole-body and kinematic analysis)”. We have analyzed two videos unrelated to string pulling, 1) in which a human is throwing and catching a ball, and 2) a mouse is clasped from tail and motion of body and hind paws is observed. Raw and tagged videos are included on the online GitHub wiki sample videos section. We have made raw and analyzed data available through the OSF website. In the manuscript Figure 14 and its supplement are included as illustrations.

Please discuss the support and maintenance. Further, the poor documentation must be improved and the associated GitHub must be organized

Please see new paragraph in retitled and updated Discussion section “Support and software maintenance, future work, and possible extensions of the toolbox and the string-pulling behavioral task”. We have updated “readme” and “Support” file on GitHub delineating installation procedure and added a wiki user manual demonstrating usage with illustrations (screen clips) of the software windows and interactions. Please visit links below.

https://github.com/samsoon-inayat/string_pulling_mouse_matlab

https://github.com/samsoon-inayat/string_pulling_mouse_matlab/wiki

https://github.com/samsoon-inayat/string_pulling_mouse_matlab/wiki/Detailed-Usage-Instructions

A Video tutorial has also been added through Youtube video.

Please explicitly spell out the advantages of your approach over related methods. In this regard, address the black-box method of "DeepLabCut" and the "Janelia Animal Part Tracker".

Please see renumbered and updated Discussion section “Comparison with artificially intelligent machine learning based software”. We discuss that our approach provides direct measures of relevant parameters to analyze string pulling behavior. Instead of just pose estimation where points are tracked, with image segmentation, regions can be tracked e.g., body whose length and tilt can be measured directly from the identified region. Subsequent analyses show White mice have larger body linear and angular speed. Our approach doesn’t require large computation power and lead time is faster because no neural network training is required which can take several days. If experimental conditions are changed, retraining of networks is usually required.

Lastly, please make the toolbox available to be downloaded in Matlab 2019b (reviewers were unable to successfully download).

We tested the software with Matlab 2018b and Matlab 2019b and have updated the Materials and methods section.

[Editors' note: further revisions were suggested prior to acceptance, as described below.]

The manuscript has been improved but there are some remaining issues that need to be addressed before acceptance, as outlined below:The reviewers would like you to please provide more discussion of the significance of these findings for an average audience. The data show how whole-body measurements could be used to reveal differences in motor kinematics between mouse strains. These analyses suggest that Swiss Webster Albino (white) mice exhibit faster movements that are more localized to the forelimb compared to black 6 mice. What is the significance of these differences? For example, the authors interpret these differences as the white mice using a "more primitive strategy" than the black mice. In addition, the white mice were considered to have "exaggerated compensatory adjustment" and "impaired individualized arm/hand movements". Please elaborate on how these conclusions were derived and what are the implications for future studies?

We have added and rearranged text (as well as added more references) in the Discussion section to improve readability and highlight the significance of findings related to the differences between White and Black mice. We have replaced the indirect terms highlighted by the reviewers with more direct terms that are backed by conclusions from quantitative analysis. For example, we have replaced “more primitive” with “different”, “exaggerated compensatory adjustment” with “more body motion”, and “impaired individualized arm/hand movements” with “are displaying an impairment in arm/hand movements”. Please see changes in the Discussion section.

We also noted a number of typos and errors in figure references. Here are some examples:– In the first paragraph of the subsection “Comparison with artificially intelligent machine learning based software”, Figure 2.11 is likely referring to Figure 12.– At the end of the first paragraph of the subsection “Robustness and limitations of image segmentation and automatic detection of body parts”, the figure is incorrectly referenced. It is likely referring to Figure 15—figure supplement 1.– The figure on Time Warping should be Figure 11—figure supplement 1, not Figure 15.

We apologize for these typos as they resulted due to a major format change of the manuscript (rearrangement of the Materials and methods, Results, and Discussion sections and different figure numbering) requested by the editorial staff at *eLife*. We have now corrected these and similar typos.